# Variation of soil organic carbon, stable isotopes and soil quality indicators across an eroding-deposition catena in an historical Spanish olive orchard

José A. Gómez[1], Gema Guzmán[2], Arsenio Toloza[3], Christian Resch[3], Roberto García-Ruíz[4], Lionel Mabit[3]

[1]Institute for Sustainable Agriculture-CSIC, Córdoba, Spain
[2]Applied Physics Dept., University of Córdoba, Spain
[3]Soil and Water Management and Crop Nutrition Laboratory, FAO/IAEA Agriculture & Biotechnology Laboratory, IAEA Laboratories Seibersdorf, Austria
[4]Animal and Plant Biology and Ecology Dept., Ecology section, Center for advance studies in olive groves and olive oils. University of Jaén, Spain

*Correspondence to*: José A. Gómez (joseagomez@ias.csic.es)

**Abstract.** This study compares the distribution of bulk soil organic carbon, its fractions (unprotected, physically, chemically and biochemically protected), available P ($P_{avail}$), organic nitrogen ($N_{org}$) and stable isotopes ($\delta$ $^{15}$N and $\delta$ $^{13}$C) signatures at four soil depths (0–10, 10–20, 20–30, 30–40 cm) between a nearby open forest reference area and an historical olive orchard (established in 1856) located in Southern Spain. In addition, these soil properties, as well as water stable aggregates ($W_{sagg}$) were contrasted at eroding and deposition areas within the olive orchard, previously determined using $^{137}$Cs. SOC stock in the olive orchard (about 40 t C ha$^{-1}$) was only 25 % of that in the forested area (about 160 t C ha$^{-1}$) at the top 40 cm of soil, and reduction was especially severe in the unprotected organic carbon. The reference and the orchard soils also showed significant differences in the $\delta$ $^{13}$C and $\delta$ $^{15}$N signals, likely due to the different vegetation composition and N dynamics in both areas. Soil properties along a catena, from erosion to deposition areas within the old olive orchard, showed large differences. Soil $C_{org}$, $P_{avail}$ and $N_{org}$ contents and $\delta$ $^{15}$N at the deposition (presumably non-degraded) were significantly higher than that of the erosion area, defining two distinct areas with a different soil quality status (non-degraded vs degraded). These overall results indicate that proper understanding of $C_{org}$ content and soil quality in olive orchards require the consideration of the spatial variability induced by erosion/deposition processes for a convenient appraisal at farm scale.

## 1 Introduction

Research on soil organic carbon (SOC) and its dynamics has increased after the declaration of 4 per thousand program (Lal, 2015), which seeks to increase global soil organic carbon stocks by 0.4 percent per year as a compensation for global anthropogenic C emissions. Under this program, special emphasis is given to combat soil degradation due to the strong

impact on the global carbon cycle because of the depletion of the SOC stock. For instance, in European agricultural soils, Lugato et al. (2016) reported that erosion-induced SOC fluxes were in the same order as the current gains from improved management and must be reduced to maintain soil health and productivity. Lal (2003) estimated the global erosion-induced

displacement of SOC at 5.7 Pg C yr$^{-1}$, approximately 70 % of which is redistributed and redeposited over the landscape and the remaining 30% is transported by rivers into aquatic ecosystems. SOC is the most important indicator of soil quality (Rajan et al., 2010) and erosion-induced loss of SOC affects on-site soil fertility and off-site environment quality (Lal, 2019). However, the effects of soil erosion and the fate of the specific SOC fraction transported by erosion in specific agricultural systems such as olive cropping remains poorly understood, and therefore, agro-environmental impacts of SOC dynamics and

variability require more site and crop specific research.

Olive trees, one of the most important crops in the Mediterranean region which account for approximately 9.7 Mha (FAOSTAT, 2019), have been linked to severe environmental impacts including the acceleration of erosion and soil degradation (e.g. Beaufoy, 2001, Scheidel and Krausmann, 2011). In fact, soil degradation is common in olive orchards as they have been traditionally cultivated under rainfed conditions on sloping land, at relatively low tree densities, limited

canopy size by pruning and bare soil management to optimize water use by the tree under the semiarid conditions which characterize the Mediterranean climate (Gómez, 2014). Indeed, there are many studies which have measured high erosion rates in olive orchards on sloping areas (e.g. Gómez et al., 2014), although these high erosion rates are not necessarily a direct consequence of current management. Vanwalleghem et al. (2011) in a study of historical erosion rates in several ancient olive orchards of Montefrío (Southern Spain) reported unsustainable erosion rates in the range of 23 to 68 Mg ha$^{-1}$ y$^{-1}$

during the XIX and early XX centuries, when these orchards were managed under the same slope and rainfall conditions with bare soil, albeit based on animal ploughing. Vanwalleghem et al. (2011) also reported a further increase in the erosion rates when bare soil management started to be implemented in these orchards by mechanization and herbicides, in the late XX century. In the last five decades (Ruíz de Castroviejo, 1969), there has been an attempt to control soil degradation, while maintaining a favourable soil water balance for the tree through the gradual development of temporary cover crops (grown

during the rainy season) (Gómez et al., 2014). These high erosion rates have also been linked to the degradation of soil properties observed in olive orchards. For instance, Gómez et al. (2009b) measured the differences in soil properties in a 5-year long experiment on runoff plots reporting a decrease in SOC, aggregate stability and infiltration rates in bare soil as compared to cover crop management. Such scientific evidence which links changes in soil properties to different erosion rates in olive orchards under controlled conditions are rarely reported in the literature. Indeed, most of the studies connecting

soil properties with different soil managements in olives come from surveys of soil properties in orchards placed on similar soil types but with differences in soil management. An example of these studies are those of Álvarez et al. (2010) or Soriano et al. (2014) who found an improvement in soil properties, particularly aggregate stability, SOC and biological activity, in organic olive orchards with cover crops, when compared to bare soil ones. In recent years, these studies have started to deepen our understanding in investigating key properties such as SOC. For instance, Vicente-Vicente et al. (2017) evaluated

the impact of cover crops in the distribution of unprotected and protected SOC in the top 15 cm of the soil. These field

studies take samples in a representative area of the slope, which is a common assumption in many soil quality studies (e.g. Andrews and Carroll, 2001). Although there are a limited number of experiments on the spatial variability of soil properties in olive orchards, they suggest significance in-field variability (e.g. Gargouri et al., 2013; Huang et al. 2017). Moreover, Gómez et al. (2012) suggested that part of this on-site variability of soil properties, regarding organic carbon, might be related to erosion/deposition processes.

In-field variability associated with erosion/deposition processes is relatively well documented for organic carbon content in field crops (e.g., De Gryze et al. 2008, Mabit and Bernard, 1998, 2010; Van Oost et al., 2005).While the human-induced acceleration of soil erosion has depleted the SOC stock of agroecosystems, the fate of SOC transported over the landscape and that deposited in depressional sites is not fully understood, despite the fact that it might explain a high proportion of the on-site variability of soil properties.

Most of the erosion rates recorded or established in olive orchards come from runoff plots or small catchment experiments (e.g. Gómez et al., 2014). The use of the $^{137}$Cs approach has demonstrated its potential in establishing long-term soil erosion rates in this specific land use. An example of these studies is that of Mabit et al. (2012) in which erosion as well as deposition rates since the 1950´s were determined in one ancient olive orchard in the municipality of Montefrío, showing an average annual rate in the eroding part of the slope of 12.3 t ha$^{-1}$ yr$^{-1}$, and an average deposition rate in the lower section of the hillslope, much shorter than the eroding section, of 13.1 t ha$^{-1}$ yr$^{-1}$. This study involved a reference area for establishing precisely the initial $^{137}$Cs inventory, a natural undisturbed area located at 200 m from the orchard. As reported by Mabit et al. (2012), based on 13 investigated soil profiles, the initial $^{137}$Cs fallout in this undisturbed area was evaluated at 1925 ± 250 Bq m$^{-2}$ (mean ± 2 standard error) with a CV of 23%.

To complement and/or to circumvent some limitation associated with the use of this anthropogenic radioisotope (see Mabit et al., 2008) and to maintain the capacity to determinate erosion and deposition rates without the need to use direct measurements, other natural radioisotopes such as $^{210}$Pb (e.g. Mabit et al., 2014; Matisoff et al., 2014) or stable isotopes such as δ $^{15}$N or δ $^{13}$C (e.g. Meusburger et al., 2013) have been proposed.

In this study, we hypothesized that the contribution of the long-term erosion-deposition processes on the in-field variability of soil properties in olive orchards (or other woody crops) under medium-high slope is relevant and should be taken into account when analysing the effects of specific strategies on SOC sequestration or on other soil properties. In addition, we exploited the advantage provided by the unique location of an ancient olive orchard near an undisturbed reference area and the previous information on this site from studies on historical erosion rates, to fulfil the following objectives:

1- To quantify the long-term variability in soil total organic carbon and in their different fractions, and soil quality indicators in relation to erosion and deposition areas in an historical olive orchard;

2- To evaluate these differences in relation to the reference values found in an undisturbed natural area;

3- To evaluate differences in stable isotopes (δ $^{13}$C and δ $^{15}$N) and explore their potential for identifying degraded areas within the olive orchard.

**2 Materials and Methods**

2.1 Description of the area

The study area is located in the municipality of Montefrío, southwestern Spain (Figure 1). The municipality extension is around 220 km$^2$, of which 81 % is cultivated, mostly with olive trees. The climate in the region is continental Mediterranean with a long-term (1960–2018) average annual precipitation of 630 mm, a mean annual evapotranspiration of 750 mm, and a yearly average temperature of 15.2 ℃. It is a mountainous area, with elevation ranging between 800–1600 m a.s.l. at the highest point (Sierra de Parapanda). Soil sampling took place in two areas around the archaeological site "Peña de los Gitanos", where the soil is classified as Calcic Cambisol according to the FAO classification. The reference undisturbed area was inside an archaeological site (Figure 1). This undisturbed area is covered by open Mediterranean forest interspersed with shrubs and annual grasses on limestone material (calcarenites). The status of this protected site guarantees that no anthropogenic activities have impacted on it for a long period of time, approximately since the end of XVI century. This area is covered by natural vegetation typical of the Mediterranean region, mainly bushes like *Pistacia lentiscus* and *Retama sphaerocarpa* and herbaceous species such as *Anthemis arvensis*, *Calendula arvensis, Borago officinalis, Bracchypodium* spp*., Bromus* spp*., and Medicago* spp. Combined with its flat topography, this area has the potential to allow the establishment of reference values for undisturbed soil. The area studied was an olive orchard located close to (some tens of meters) the reference area (Figure 1) which had been established in 1856. Both areas were described in detail in previous studies on historical erosion rates in the region (Vanwalleghem et al., 2011, Mabit et al., 2012). This olive orchard is rainfed, and soil management in the decades before the sampling was based on bare soil with pruning residues (trees pruned every 2 years) being chopped and left on the soil surface. Olive trees are fertilized annually with 5 kg of 15 N-P-K, spread below the tree canopy area.

2.2 Soil sampling

The reference site, adjacent to the olive orchard, belongs to an archaeologically protected site and therefore it is a non-cropped area excluded from any soil disturbance, Figure 1. This site was sampled at thirteen points across a transect, spaced at an average distance of 6 m, with only four of these sampling points used in this study (Figure 1). At each sampling point, the excavation method was used and based on the collection of soil samples at 5 cm increments until bedrock was reached (i.e. 0–5, 5–10, 10–15, 15–20 and when possible, 20–25, 25–30, 30–35, 35–40, 40–45, 45–50, 50–55 and 55–60 cm). Composite soil samples at 10 cm interval were prepared at the laboratory to perform the chemical analysis of the reference area as described below.

In the olive orchard a hydraulic mechanical core sampler was used. It gently rotates and pushes an 8 cm in diameter core to sample 8 points in a 452 m long catena (Figure 1). To minimize soil disturbance, soil sampling was made in soil water content between 40 to 80 % of water holding capacity. Precautions were taken to assure that whole sample was collected for

each given depth, and at each sampling point in the orchard soil was taken at four different depths (0–10, 10–20, 20–30 and 30–40 cm).

In a previous study, soil erosion and deposition rates were determined at each sampling point, comparing the [137]Cs inventory among these points and that of the undisturbed reference area (Mabit et al., 2012). The positions of all sampling points were recorded by RTK-GPS at submeter resolution (Table 1). Overall, 12 points were sampled, 4 in the reference area and 8 in the

135 catena across the olive orchard, with all of them reaching the bedrock below 40 cm depth.

### 2.3 Physicochemical analysis

Soil samples were passed through a 2 mm sieve and homogenized, and stoniness determined as % in mass. Separation of the various soil organic carbon ($C_{org}$) pools was performed by a combination of physical and chemical fractionation techniques through a three-step process developed by Six et al. (2002) and modified by Stewart et al. (2009), summarized here. This

three-step process isolates a total of 12 fractions and it is based on the assumed link between the isolated fractions and the protection mechanisms involved in the stabilization of organic C. First a partial dispersion and physical fractionation of the soil is performed to obtain three size fractions: >250 mm (coarse non-protected particulate organic matter, POM), 53–250 mm (microaggregate fraction), and <53 mm (easily dispersed silt and clay). This physical fractionation is done on air-dried 2-mm soil sieved over a 250-mm sieve. Material greater than 250 mm remained on the sieve. Microaggregates were

collected on a 53-mm sieve that was subsequently wet-sieved to separate the easily dispersed silt- and clay-sized fractions from the water-stable microaggregates. The suspension was centrifuged at 127 x g for 7 min to separate the silt-sized fraction. This supernatant was subsequently separated, flocculated and centrifuged at 1730 x g for 15 min to separate the clay-sized fraction. All fractions were dried in a 60 ºC oven and weighed. Afterwards there was a second step involving a further fractionation of the microaggregate fraction isolated in the first step. A density flotation with sodium polytungstate

was used to isolate fine non-protected POM (LF): After removing the fine non-protected POM, the heavy fraction was dispersed overnight by shaking and passed through a 53 mm sieve to separate the microaggregate-protected POM (>53 mm in size, iPOM) from the microaggregate-derived silt and clay-sized fraction. The resulting suspension was centrifuged to separate the microaggregate-derived silt- versus clay-sized fraction as described above. A final third step involved the acid hydrolysis of each of the isolated silt- and clay-sized fractions. The silt- and clay-sized fractions from both the density

flotation and the initial dispersion and physical fractionation were subjected to acid hydrolysis. The unprotected pool includes the POM and LF fractions, isolated in the first and second fractionation steps, respectively. The physically protected SOC consists of the SOC measured in the microaggregates. It includes not only the iPOM but also the hydrolysable and non-hydrolysable SOC of the intermediate fraction (53–250 μm). The chemically and biochemically protected pools correspond to the hydrolysable and non-hydrolysable SOC in the fine fraction (< 53 μm), respectively. In all cases, SOC fractions and in

the bulk soil, organic carbon concentrations were determined by using the wet oxidation sulfuric acid and potassium dichromate method of Anderson and Ingram (1993).

Inorganic carbon was removed prior to stable isotope analysis by acid fumigation following the method of Harris et al. (2001). Moistened subsamples were exposed to the exhalation of HCl in a desiccator overnight. Afterwards, the samples were dried at 40º C before measuring the stable isotope ratio. The N measurements were done with unacidified samples and the stable N isotope ratios and the C and N concentrations were measured by isotope ratio mass spectrometry (Isoprime 100 coupled with an Elementar Vario Isotope Select elemental analyser; both instruments supplied by Elementar, Langenselbold, Germany). The instrumental standard deviation for $\delta$ $^{15}$N is 0.16% and 0.11% for $\delta$ $^{13}$C. Stable isotopes are reported as delta values (º/$_{oo}$) which are the relative differences between the isotope ratios of the samples and the isotope ratio of a reference standard.

In addition, available phosphorus ($P_{avail}$) was determined by the Olsen method (Olsen and Summers 1982) and organic nitrogen ($N_{org}$) was determined by the Kjeldahl method (Stevenson, 1982). Water stable aggregates ($W_{sagg}$) were measured using the method of Barthes and Roose (2002). Soil particle size distribution was determined using the hydrometer method (Bouyoucos, 1962) for the topsoil (0–10 cm) of the reference area and the olive orchard. Bulk density values were calculated for the whole profile based on the volume, soil (material finer than 2 mm) and stone contents determined from the excavation and core sampling described above. Additionally, top soil (0-10 cm depth) soil bulk density was measured using a manual soil core sample with a volume of 100 cm$^3$. Soil carbon stocks were calculated for the fine soil fraction after discounting rock or stone fragments larger than 2 mm, and considering bulk density, and presented on equivalent soil mass as described by Wendt and Hauser (2013). As proposed by Hassink and Whitmore (1997), theoretical values of carbon saturation were calculated from the soil particle analysis. Finally, the soil degradation index developed by Gómez et al. (2009a) was calculated from the $C_{org}$, $P_{avail}$ and $W_{sagg}$.

### 2.4 Statistical analysis

The overall effect of depth and area (reference site vs. olive orchard or eroded vs. deposition area within the olive orchard) were evaluated using a two factor ANOVA ($p<0.05$). Additionally, for some comparison at similar soil depth, values of soil properties between two different areas were assessed using a one-way ANOVA test ($p<0.05$). In both situations, data were log-transformed when necessary to fulfil ANOVA requirements. Exploratory analysis using Principal Component Analysis (PCA) was performed in the olive orchard area using the variables and sampling depths showing significant differences in the ANOVA analysis. This PCA analysis was complemented by determining the linear correlation coefficient variables showing the highest load on the PCA analysis and erosion/deposition rates using the Pearson correlation test. The statistical software package Stata SE14.1 was used for these analyses.

### 3 Results

### 3.1 Organic carbon concentration and distribution among fractions

Table 2 shows the significance of the differences in bulk soil $C_{org}$ and the various $C_{org}$ fractions between reference and olive orchard plots, between soil depth and due to the interaction between both (Table 2A), and the effects of erosion/deposition ratio (Table 2B). Bulk soil $C_{org}$ is always significantly higher in the reference area as compared to the olive orchard (Table 2A and Figure 2), and this was independent of the soil sampling depth. $C_{org}$ values on the reference site were between 2 to 5 times higher than that of the olive orchard for a given depth, with the greater differences in the top 10 cm of the soil. Soil depth has a significant effect on bulk $C_{org}$ and $C_{org}$ fractions, with values typically decreasing with depth in both areas. $C_{org}$ concentrations in the unprotected, and physically, chemically and biochemically protected fractions were significantly higher in the reference site as compared to the olive orchard (Table 2A and Figure 3). $C_{org}$ values were between 2 to 6 times higher for the unprotected and chemically protected fractions, and between 2 to 3.5 times for the physically and biochemically protected fractions, with differences tending to decrease with the soil depth.

Within the olive orchard, there were statistically significant differences between the erosion and deposition areas (Table 2B). Higher $C_{org}$ values (1.1 to 0.6%) were observed in the deposition area located downslope, whereas lower values (0.85 to 0.55 %) were measured in the areas with net erosion in the upper and mid sections of the catena. It is worth noting that these differences between erosion and deposition areas are detected for overall analysis using a two-way ANOVA (Tables 2A and B), although an individual analysis at each depth (Figure 2) does not detect statistically significant differences, probably due to the moderate number of replications. Significant differences between the deposition and eroding area were also found for the unprotected and the physically and chemically protected fractions (Table 2B, Figure 4). However, differences for the biochemically protected (Table 2B, Figure 4) were not significant.

The percentage distribution of SOC among fractions was also significantly different between both areas (reference vs. olive orchard), except for the biochemically protected fraction (Table 3A, Figure 5). The reference area lays up most of the $C_{org}$ in the unprotected fraction (between 50 and 65% approximately) with no significant trend with depth (Table 3A, Figure 5), followed in relative importance by the chemically and physically protected fractions which contributed between 18–30 % and 10–20 % of the bulk soil $C_{org}$, respectively. The biochemically protected fraction represents a very low percentage (between 4 to 6 %). In the olive orchard, $C_{org}$ is stored predominantly in the physically and chemically protected fractions which accounted for about 38 to 27 and 34 to 28 % respectively, followed by the pool of unprotected fraction (between 22 to 32%) (Figure 5). The biochemically protected fraction represents from 11 to 4% of the organic carbon stored in the olive orchard. There are no clear differences in the organic carbon distribution among the different fractions between the erosion and deposition areas in the olive orchard, with the exception of the physically protected fractions at 10-20 cm depth (Table 3B and Figure 6).

### 3.2. Organic carbon stock

SOC stock in the reference area is approximately 160 t ha$^{-1}$ being significantly higher than that of the olive orchard for an equivalent mass (Figure 7), which stores between 38 and 41 t ha$^{-1}$ in the eroded and deposition areas respectively. There were no significant differences between these two orchard areas. Similar results were achieved across the top 40 cm soil

profile. Clay, silt and sand contents of the topsoil (0–10 cm) along the catena in the olive orchard averaged 41, 37 and 22%, respectively. Variability was low (average coefficient of variation of 17%) and there were no significant changes between the erosion and deposition areas. In the reference area, the soil has an average clay, silt and sand content of 30, 31 and 39% respectively, also with a homogeneous distribution across the sampling area (coefficient of variation of 10%). According to the Hassink and Whitmore (1997) model, the percentages of organic carbon of maximum soil stable $C_{org}$ are of 3.24±0.11

and 3.63±0.19 % in the reference site and olive orchard, respectively. So, protected $C_{org}$ in the reference and olive orchard areas account for 49.8±11.5 % and 20.49±5.2 % of the maximum soil stable $C_{org}$, respectively, in the topsoil.

## 3.3. δ $^{15}$N and δ $^{13}$C isotopic signal

Figure 8 and Table 4A compare stable isotope delta values between the reference site and the overall olive orchard by depth. There are significant statistical differences in δ $^{15}$N, δ $^{13}$C, and δ $^{13}$C:δ $^{15}$N ratio between the two areas, although in the case

of δ $^{15}$N only in the top 20 cm. Soil depth had a significant effect. When comparing differences between the erosion and deposition areas within the olive orchard, we detected statistical significant differences only in δ $^{15}$N and δ $^{13}$C:δ $^{15}$N ratio, most marked in the top 20 cm of the soil (Figure 8, Table 4B).

## 3.4. Soil quality of topsoil across the catena

Figure 9 depicts the comparison between the $P_{avail}$, $N_{org}$, $W_{sagg}$ as well as the soil degradation index (SDI, Gómez et al.

2009a) at the top 10 cm of the soil between the erosion and deposition areas of the olive orchard and Table 5 shows a similar comparison for $N_{org}$, $P_{avail}$ and bulk density at the different soil depths. $P_{avail}$ in the deposition area is much higher than that of the erosion area in the top soil and for the whole profile, whereas no significant differences in individual soil layers were found for $N_{org}$ and $W_{sagg}$. SDI, which is an aggregated indicator of these three soil variables, in the eroded area it was about 3 times higher than that in the deposition area.

## 3.5. Overall analysis of soil property variability between eroded and deposition areas in the orchard

Table 6 shows the loads in the three first principal components (PC) of the principal component analysis (PCA) for the variables used in this analysis. More than 70% of the variance was explained by the first two PC. Soils of the eroded and deposition area were clearly separated in the space defined by the two PC (Figure 10). The variables with higher contributions in PCs 1 and 2 were related to $P_{avail}$ concentration, measured in the in 0- 10 cm and also on the 0- 40 cm of the

soil profile, to δ $^{15}$N, δ $^{13}$C on 10 to 30 cm soil depth, and to $C_{org}$ concentration and distribution in some fractions also in soil depths between 10 to 30 cm (Table 6). Deposition area tended to show higher values in the PC1 and PC2, and there was no clear tendency in the erosion area along the catena. The linear correlation coefficient between variables and the erosion/deposition rate was rather significant, r > ±0.742 for most of the variables. Interestingly, $P_{avail}$ in the whole soil layer was highly positively correlated. Figure 11 depicts the ones with the clearest correlation with erosion/deposition rates, being

the two most robust correlations with $P_{avail}$ concentration across the 40cm soil depth, which presented a positive correlation with deposition rates, and $\delta$ $^{13}$C at 10-20 cm depth which showed a negative correlation with deposition rates.

## 4 Discussion

### 4.1 Organic carbon concentration and distribution among fractions

After approximately 175 years of contrasted land use between the undisturbed reference site and the olive orchard, bulk soil

organic carbon concentration and its fractions have been dramatically reduced in the olive orchard. Current levels of $C_{org}$ concentration in the soil profile are approximately 20–25% of the reference area covered by the natural vegetation in the area adjacent to the orchard. This ratio is similar, albeit in the lower range, of the comparison of $C_{org}$ in topsoil among olive orchards with different managements and natural areas reported for the region (Millgroom et al., 2007). The increased soil disturbance, the lower annual rate of biomass returned to the soil and the higher erosion rate in the olive orchard explain this

difference. In both areas, the $C_{org}$ is clearly stratified, indicating that despite the different mechanisms involved there is a periodic input of biomass from the olive trees (e.g. fall of senescence leaves and tree pruning residues) plus the annual ground vegetation. Vicente-Vicente et al. (2017) estimated this biomass contribution in the range of 1.48 to 0.56 t ha$^{-1}$ yr$^{-1}$. It is worth noticing that the decrease in $C_{org}$ as compared to the natural area is much higher than the reported rates of increase in $C_{org}$ in olive orchards using conservation agriculture (CA) techniques, such as cover crops and incorporation of organic

residues from different sources. In a meta-analysis Vicente-Vicente et al. (2016) found a response ratio (the ratio of $C_{org}$ under CA management as compared to bare soil managed orchard) from 1.1 to 1.9 suggesting that under CA management, which combines cover crops and organic residues, $C_{org}$ doubled as a maximum.

Combining all $C_{org}$ data of the olive orchard, the variability was about 35% which is similar to what has been reported so far in the few studies found on soil $C_{org}$ variability in olive orchards. For instance, Gargouri et al. (2013) indicated a 24%

coefficient of variation (CV) in a 34 ha olive orchard in Tunisia, while Huang et al. (2017) reported an average CV of 41% in a 6.2 ha olive orchard in Southern Spain. Neither of these two studies reported clear trends in the distribution of $C_{org}$ with topography. Huang et al. (2017) pointed out the additional difficulties in the determination of $C_{org}$ due to the topography heterogeneity, although this was compounded by the fact that within the orchards there were two areas with different planting dates for the trees. Gómez et al. (2012) reported a CV of 49% with higher $C_{org}$ in areas where there was a change in

the slope gradient from the hillslope to a draining central channel into the catchment, although they could not find a simple relationship between the increase in content and the topographic indexes. Despite the fact that a lot of work has been done on the correlation between erosion-deposition and the redistribution of soil $C_{org}$, (e.g. Van Oost et al. 2005), our study is, to our knowledge, the first attempt to quantify this in detail under olive orchard agro-environmental conditions. The variability induced by the combined effects of water and tillage erosion in this olive orchard was similar to that described in other

agroecosystems. For instance, Van Oost et al. (2005) measured on two field crop sites under temperate climate, a clear correlation between the erosion-deposition rates and the topsoil $C_{org}$ concentration, which ranged between 0.8 % of the

erosion to 1.4% of the deposition sites in the top 25 cm of the soil. Besides this, Bameri et al. (2015), in a field crop site with a semi-arid environment, also measured a higher $C_{org}$ in the lower part of the field where deposition of the eroded soil from the upper zones took place with a mean $C_{org}$ value of 0.95% in the top 20 cm of the soil and a CV of 53%. Overall, under

such landscapes cultivated for a long time, the cumulative effect of tillage and water erosion on the redistribution of soil across the slope has been observed (Dlugoß et al., 2012). These processes also produce a vertical redistribution of $C_{org}$ resulting in a relatively homogeneous profile in the tilled layer (top 15–20 cm) and a gradual decline below this depth, as noted in our study.

**4.2. Organic carbon stock**

The differences in soil organic carbon stock between the reference site and the olive orchard are similar to those described previously when comparing cropland and forested areas, with the latter presenting a higher concentration of $C_{org}$, and most of it in the unprotected fraction, while the cropland presented a higher fraction of the carbon in the physically and chemically protected fractions (e.g. Poeplau and Don, 2013). This is likely due to the fact that under soil degradation processes, such as

water erosion, and low annual organic carbon inputs, as is the case under olive orchard land use, most of the unprotected $C_{org}$ decomposes relatively quickly and a great proportion of the remained low SOC is protected. In addition, the mobilisation of the unprotected $C_{org}$ is expected to be reduced in the protected forested area because of the canopy and the existing vegetation on the ground that protects the soil against runoff and splash erosion processes. In fact, the protected $C_{org}$ concentration in the topsoil of the olive orchard in the eroded area is about the 18.6±3.9 % of the upper limit of protected

$C_{org}$ (3.64±0.23 %) according to the model of Hassink and Whitmore (1997). Therefore, the low unprotected SOC concentration found in the olive orchard is an issue in the increase of SOC stocks. This is because protected fractions are fuelled from recently derived, partially decomposed plant residues together with microbial and micro, meso and macrofaunal debris (unprotected organic carbon) throughout processes like SOC aggregation into macro- and/or microaggregates (physically protected SOC) and complex SOC associations with clay and silt particles (chemically protected SOC) which are

disrupted in the cropland area in comparison with the reference area. The distribution among soil $C_{org}$ fractions in the orchard of this study was similar to the result obtained by Vicente-Vicente et al. (2017) who measured $C_{org}$ fractions distribution in olive oil orchards with temporary cover crops, with the exception of the unprotected SOC, which was much higher in soils under cover crops than that of our study under bare soil. Also interesting is the difficulty obtaining statistically significant differences in SOC stock between the eroding and deposition area (Figure 7) despite the apparent clear differences between

the two areas in some other soil properties, such as $P_{avail}$ or $\delta\ ^{15}N$ and $\delta\ ^{13}C$ isotopic signal in the subsoil (see below).

**4.3. $\delta\ ^{15}N$ and $\delta\ ^{13}C$ isotopic signal**

Differences in vegetation types induced differences in $\delta^{13}C$ between the olive orchard and the reference area but, as expected, no differences in $\delta^{13}C$ were detected between the erosion and deposition areas in the olive orchard given the same

origin of vegetation derived organic matter, C3-plants. Interestingly, within the olive orchard, significant differences in $\delta^{15}N$ were detected between the erosion and deposition areas, especially for the top 20 cm of soils (Figure 9). This suggests the potential of using $\delta^{15}N$ as a variable for identifying degraded area in olive growing fields, as has been proposed for other eroding regions in the world (e.g. Meusburger et al. 2013), which might provide an alternative when other conventional or isotopic techniques are not available. Nevertheless, further studies exploring this potential are necessary in order to also

consider the influence of the N-P fertilizer modifying the $\delta^{15}N$ in relation to the reference area (e.g. Bateman and Kelly, 2007). The source of N in soil is multifarious and subject to a wide range of transformations that affect $\delta^{15}N$ signature and therefore we can only speculate on the reasons for this difference in $\delta^{15}N$ in a relatively homogeneous area. Bulk soil $\delta^{15}N$ tended to be more positive (e.g. more enriched in $\delta^{15}N$) as N cycling rate increases soil microbial processes (e.g. N mineralization, nitrification and denitrification) resulting in products (e.g. nitrate, $N_2O$, $N_2$, $NH_3$) depleted in $^{15}N$ while the

substrate from which they were formed becomes slightly enriched (Robison, 2001). The higher $\delta^{15}N$ signature of the soil at deposition location suggests that rates of processes involved in the N cycling are higher than in the erosion area and that is in accordance with the higher bulk $C_{org}$ and $P_{avail}$ contents and lower SDI of the deposition site. The relatively lower soil $\delta^{15}N$ signature at the reference site could be partially due to the input of litter N from the natural legumes and to the closed N cycling which characterize natural forest ecosystems. The trend in $^{15}N$ enrichment with soil depth, as found in the reference

site, is a common observation in forest and grassland sites, which has been related to different mechanisms, including $^{15}N$ isotope discrimination during microbial N transformations, differential preservation of $^{15}N$-enriched soil organic matter components during N decomposition, and more recently, to the build-up of microbial $^{15}N$-enriched microbial necro mass (Huygens et al., 2008). However, there still remains the need for a careful calibration against an undisturbed reference site and a better understanding of the influence of different vegetation between the reference and the studied area in the change

of the $\delta^{15}N$ signal for its further use as an additional tool to determine soil degradation.

## 4.4. Soil quality of topsoil across the catena

This horizontal distribution due to tillage and water erosion also simultaneously affected other soil properties and has been described previously in other field crops areas. For instance, De Gryze et al. (2008) described, in a field crop area under

conventional tillage in Belgium, how $P_{avail}$ almost doubled (22.9 vs. 12.2 mg $kg^{-1}$) in the depositional area as compared to the eroding upper part. They also reported that in half of the field under conservation tillage, these differences in $P_{avail}$ between the upper and lower areas of the field disappeared. We have observed in our sampled orchard a pronounced increase in topsoil $P_{avail}$ in the deposition area of around 400% as compared to the eroding part of the orchard, which can be attributed to the deposition of enriched sediment coming from the upslope area. The cumulative effects of the differences in $C_{org}$, $P_{avail}$,

and the trend towards higher, although non-significant, $W_{sagg}$ in the deposition area, resulted in two areas within the orchard

with marked differences in soil quality: the eroded part which is within the range considered as degraded in the region (Gómez et al., (2009a)) and the depositional area, representing 20% of the orchard transect length (Table 1), which is within the range of non-degraded according to the same index. Topography and sediment redistribution by erosive processes introduce a gradient of spatial variability that questions the concept of representative area when it comes to describing a whole field. In fact, several studies (e.g. Dell and Sharpley, 2006), have suggested that the verification of compliance of environmental programs such as those related to $C_{org}$ sequestration should be based preferentially, at least partially, on modelling approaches. Our results raise the need for a careful delineation of sub-areas when analysing soil quality indicators and/or SOC carbon stock within the same field unit. It also warns about the interpretation of these delineated areas in relation to specific soil properties, avoiding over simplification of processes. For instance, in our study case erosion/deposition processes had a major impact on $P_{avail}$ and soil quality, but the impact on $C_{org}$ concentration and stock was moderate and extremely difficult to detect statistically using a moderate number of samples.

### 4.5. Overall analysis of soil property variability between eroded and deposition areas in the orchard

Our PCA and regression analysis confirmed the relatively high variability of $C_{org}$ and stock in relation to other soil quality indicators related to erosion/deposition processes, such as $P_{avail}$, commented in the previous section. While in the catena studied in this manuscript P cycle seems to be mostly driven by sediment mobilization, the $C_{org}$ and N cycle seems to be much more complex. The moderate differences in $C_{org}$ and the homogeneity in $\delta\ ^{15}N$ and $\delta\ ^{13}C$ isotopic signals between the eroding and deposition areas can be due to several processes, some of them discussed above, such as spatial variability of carbon inputs due to biomass in the plot, surface soil operations in the orchard and fertilization. We found both interesting and worth exploring in future studies that significant correlations between erosion/deposition rates and $C_{org}$, $\delta\ ^{15}N$ and $\delta\ ^{13}C$ related variables were found for samples from 10-20 and 20-30 cm, indicating that short term disturbance by surface processes can mask experimental determination of the impact of erosion deposition processes in this olive orchard for these variables.

### 5 Conclusions

1- The results indicate that erosion and deposition within the investigated old olive orchard have created a significant difference in soil properties along a catena, which is translated into different soil $C_{org}$, $P_{avail}$ and $N_{org}$ contents and $\delta\ ^{15}N$, and thus soil quality status.

2- This variability was smaller than that of the natural area, which indicated a severe depletion of SOC as compared to the natural area and a redistribution of available organic carbon among the different SOC fractions.

3- The results suggest that $\delta\ ^{15}N$ has the potential for being used as an indicator of soil degradation, although more investigation under different agroecosystems would be required for confirming this statement at a larger scale.

4- This research highlights that proper understanding and management of soil quality and $C_{org}$ content in olive orchards require considering the on-site spatial variability induced by soil erosion/deposition processes.

**6 Acknowledgements**

The authors would like to thank professor Manuel González de Molina for providing valuable insight into the management of the olive orchard in the decades before sampling. Also, in particular, to the three anonymous reviewers of the manuscript version in Soil Discussion, for their care and attention to the manuscript. All their comments and indications have greatly helped to increase the quality of our work. This study was supported by the projects AGL2015-40128-C03-01 of the Spanish
Government, SHui (GA 773903) of the European Commission and EU–FEDER funds.

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

**Figure 1: Site location and associated sampling. Left: Location map of the sampling area in Montefrío, Southern Spain. Reference site limited by the white line within a protected archaeological site (yellow line). Yellow markers in the upper right figure indicate the sampled transect within the olive orchard. Numbering starts in the points at higher elevation, see lower left figure for the elevation change in the transect in the orchard. Yellow markers in the lower left figure indicate sampling point in the reference area. Europe map designed by Freepik, and air images source Google Earth (© Google 2018).**

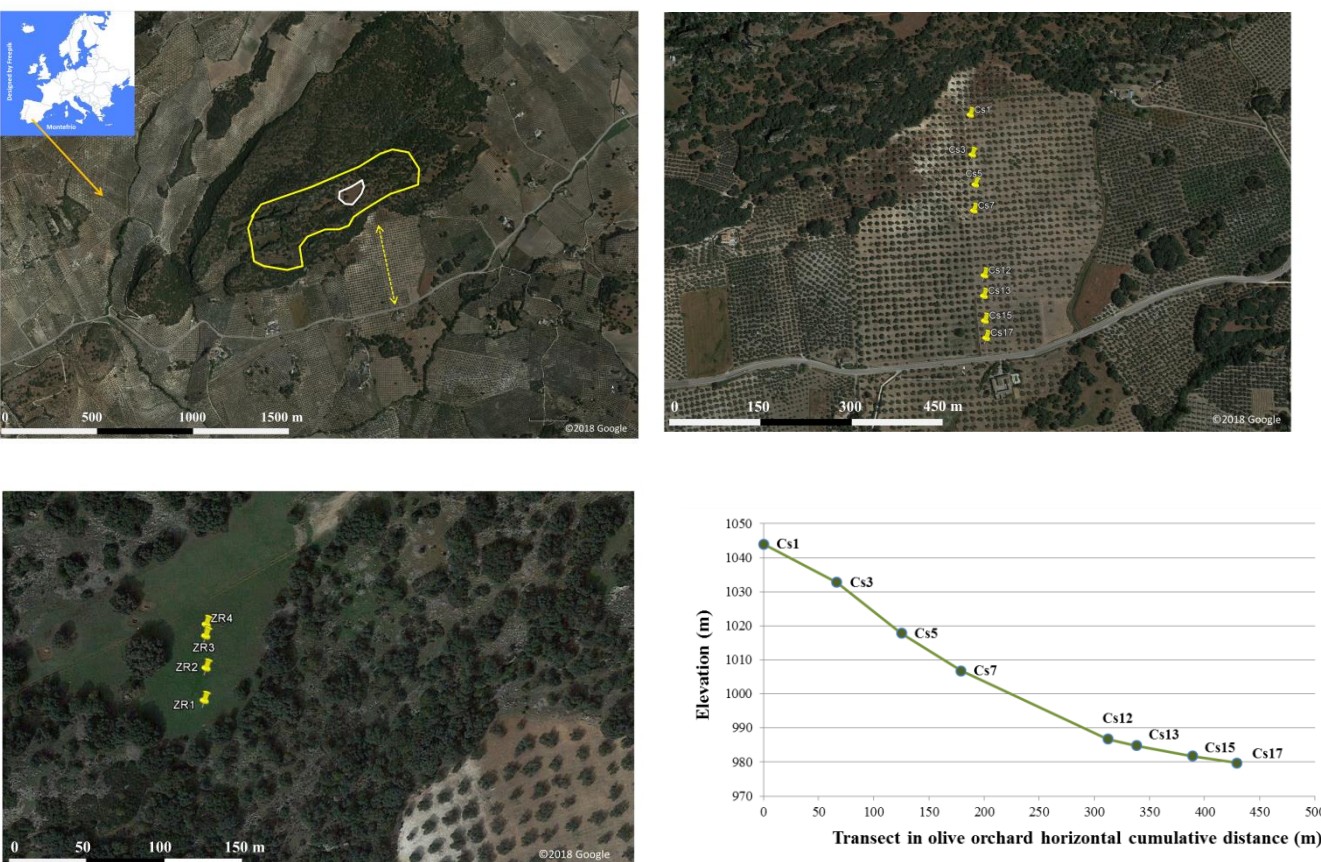

**Figure 2: Comparison of average soil organic carbon concentration in bulk soil by soil depth distinguishing among reference site, the whole olive orchard, the eroded area of the olive orchard and the deposition area in this orchard. Labels in bars for each depth indicate the p-value according to a one-way ANOVA between: a) reference vs. whole olive orchard; b) eroding vs. deposition area (lower label in *italics*).**

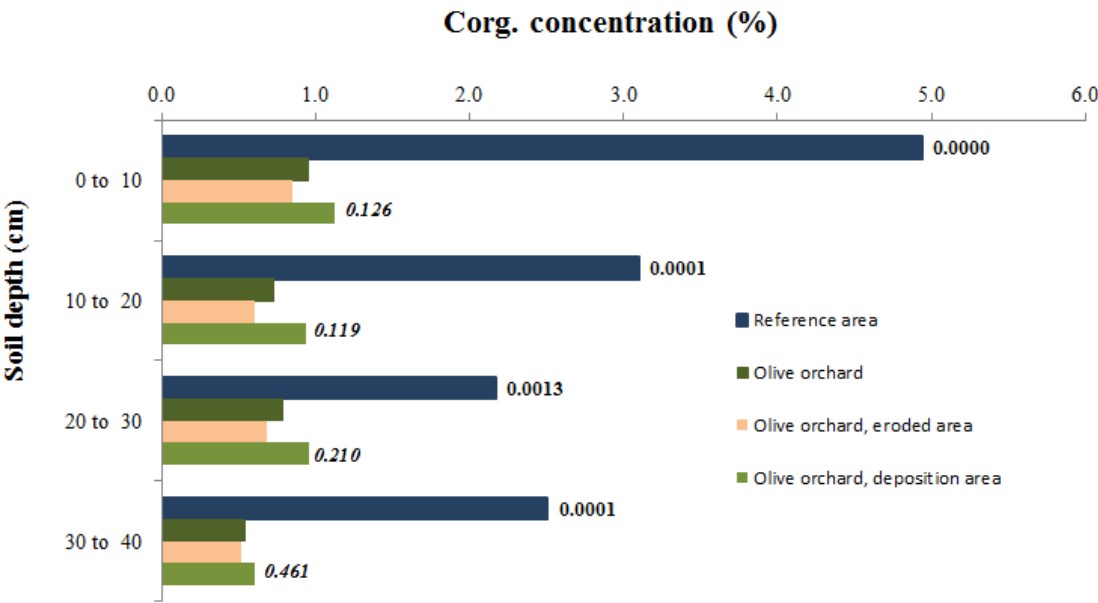

**Figure 3: Organic carbon concentration in the different soil organic carbon fractions at each depth comparing reference site vs. olive orchard. Labels in bars indicate the p-value according to a one-way ANOVA comparing treatments for the same soil depth and carbon fraction between reference area and olive orchard.**

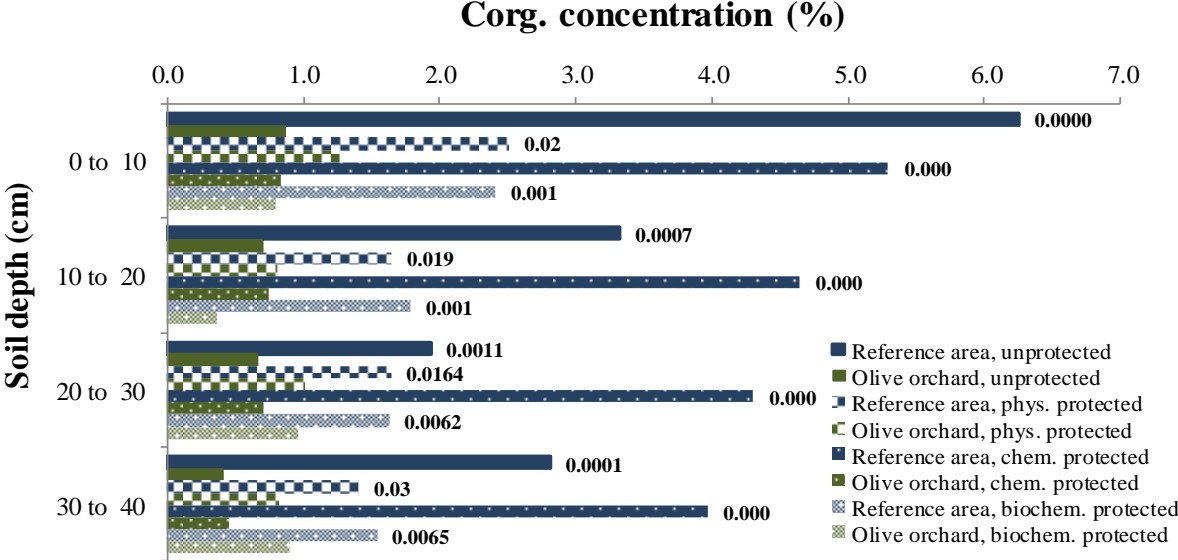

**Figure 4: Organic carbon concentration in the different soil organic carbon fractions at each depth comparing eroded vs. versus deposition area within the olive orchard. Labels in bars indicate the p-value according to a one-way ANOVA comparing treatments for the same soil depth and carbon fraction between reference area and olive orchard.**

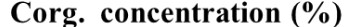

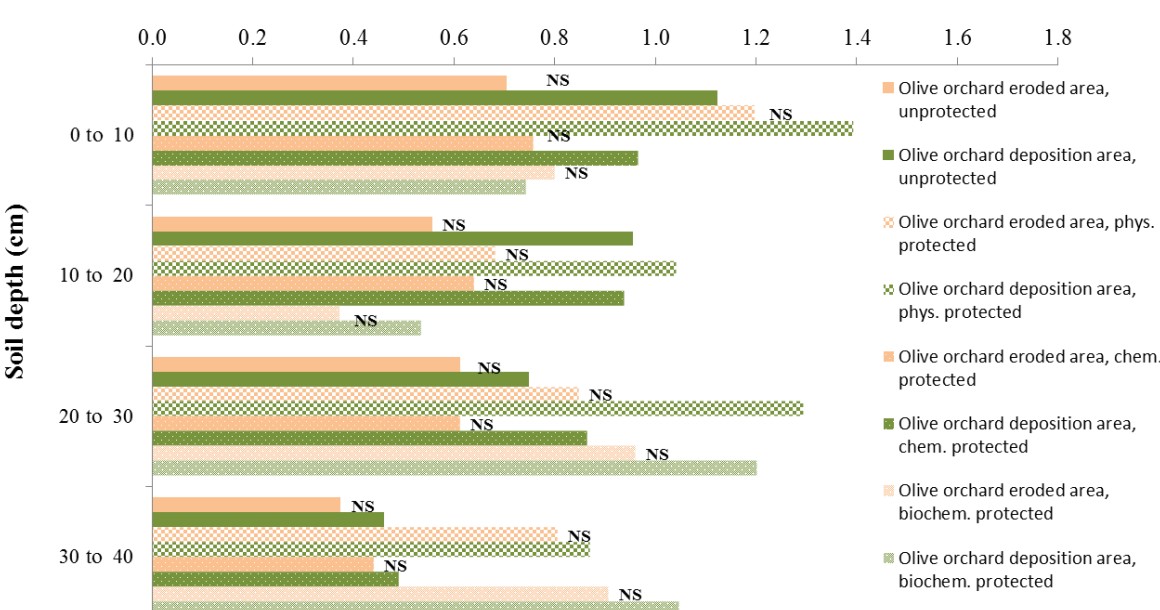

**Figure 5: Contribution (%) of the different fractions with respect to total soil organic carbon by depth comparing reference site vs. olive orchard. Labels in each fraction and depth are the p-value according to a one-way ANOVA comparing between reference area vs, olive orchard for the same carbon fraction and soil depth.**

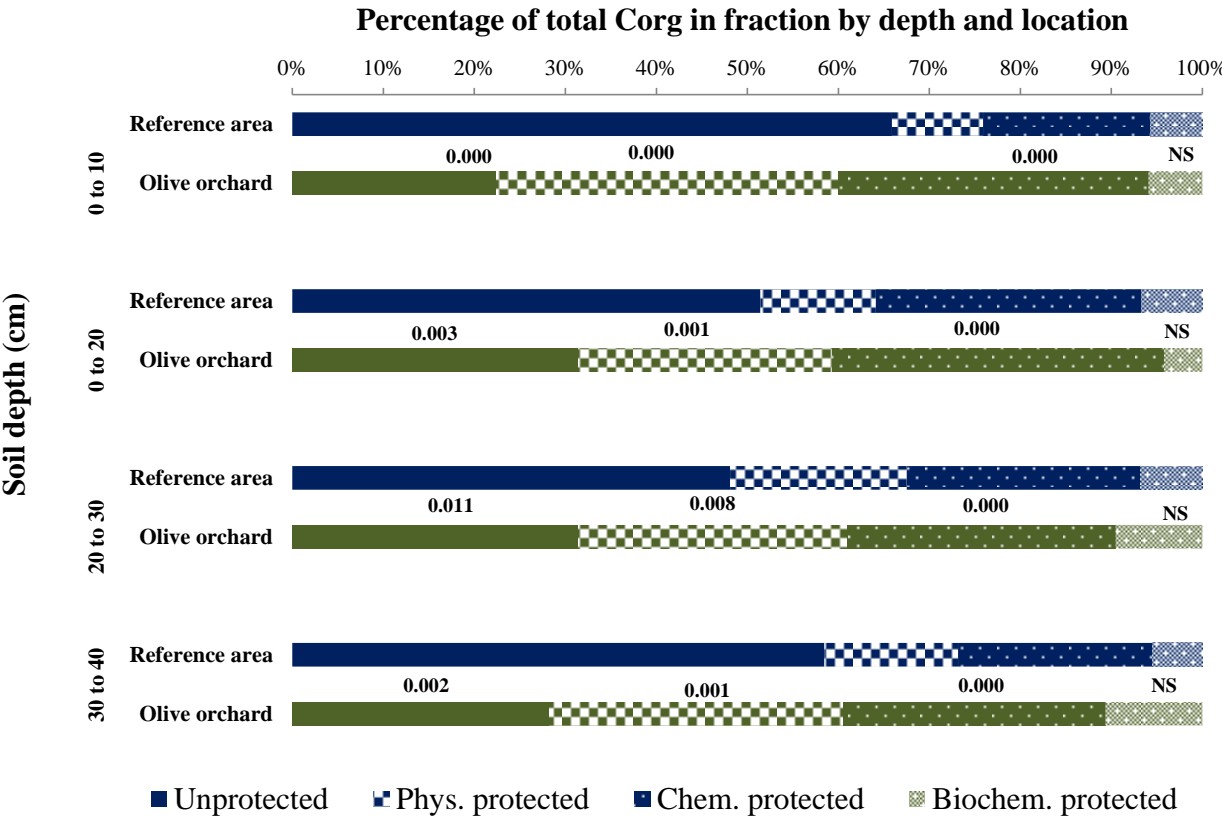

**Figure 6: Fraction of total organic carbon stored in the different fractions by depth comparing reference site vs. olive orchard. Labels in each fraction and depth are the p-value according to a one-way ANOVA comparing between reference area vs, olive orchard for the same carbon fraction and soil depth.**

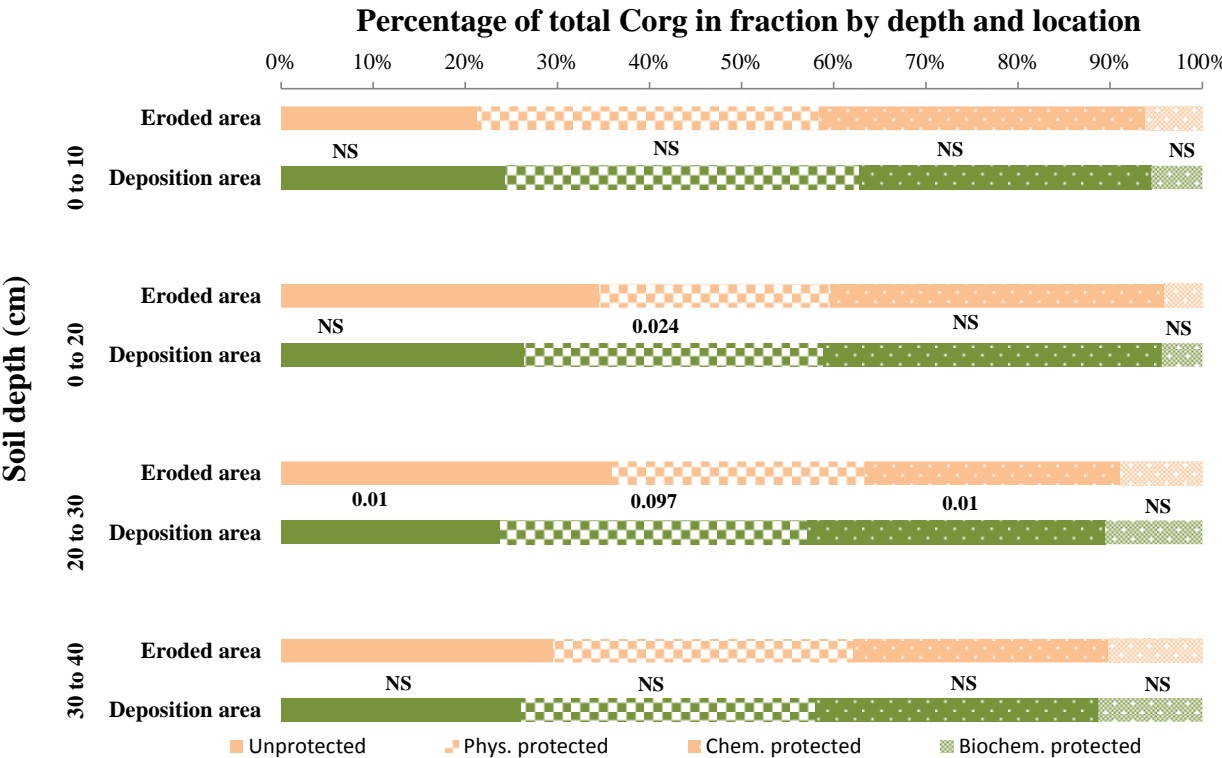

**Figure 7: Cumulative soil organic carbon (SOC) stock across the soil profile referred in cumulative soil mass on the vertical axis. Different letters for similar soil mass means statistically significant differences (Kruskal-Wallis test at p<0.05). For this analysis cumulative soil organic carbons were interpolated linearly to the average cumulative soil mass corresponding to all the points in the three areas.**

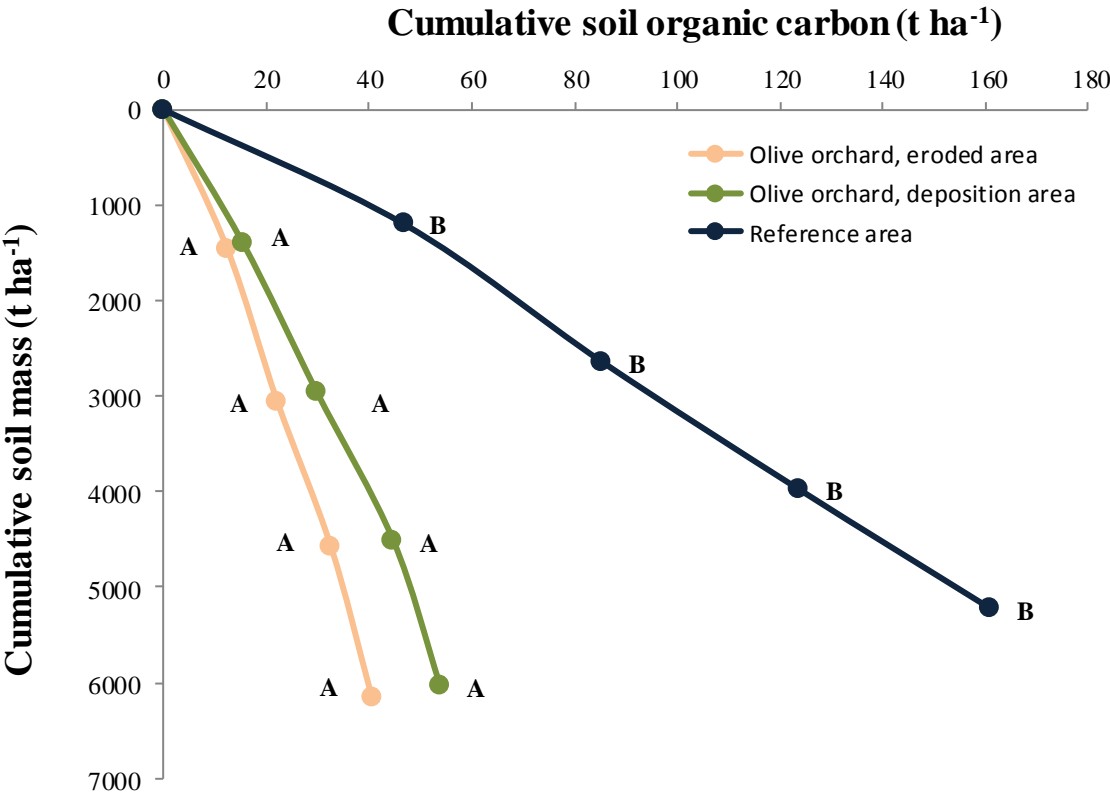

**Figure 8:** ¹³C and ¹⁵N isotopic signal of soil by depth distinguishing among reference site, the whole olive orchard, the eroded area of the olive orchard and the deposition area in this orchard. Labels in bars for each depth indicate the p-value according to a one-way ANOVA between: a) reference vs. whole olive orchard; b) eroding vs. deposition area (lower label in italics).

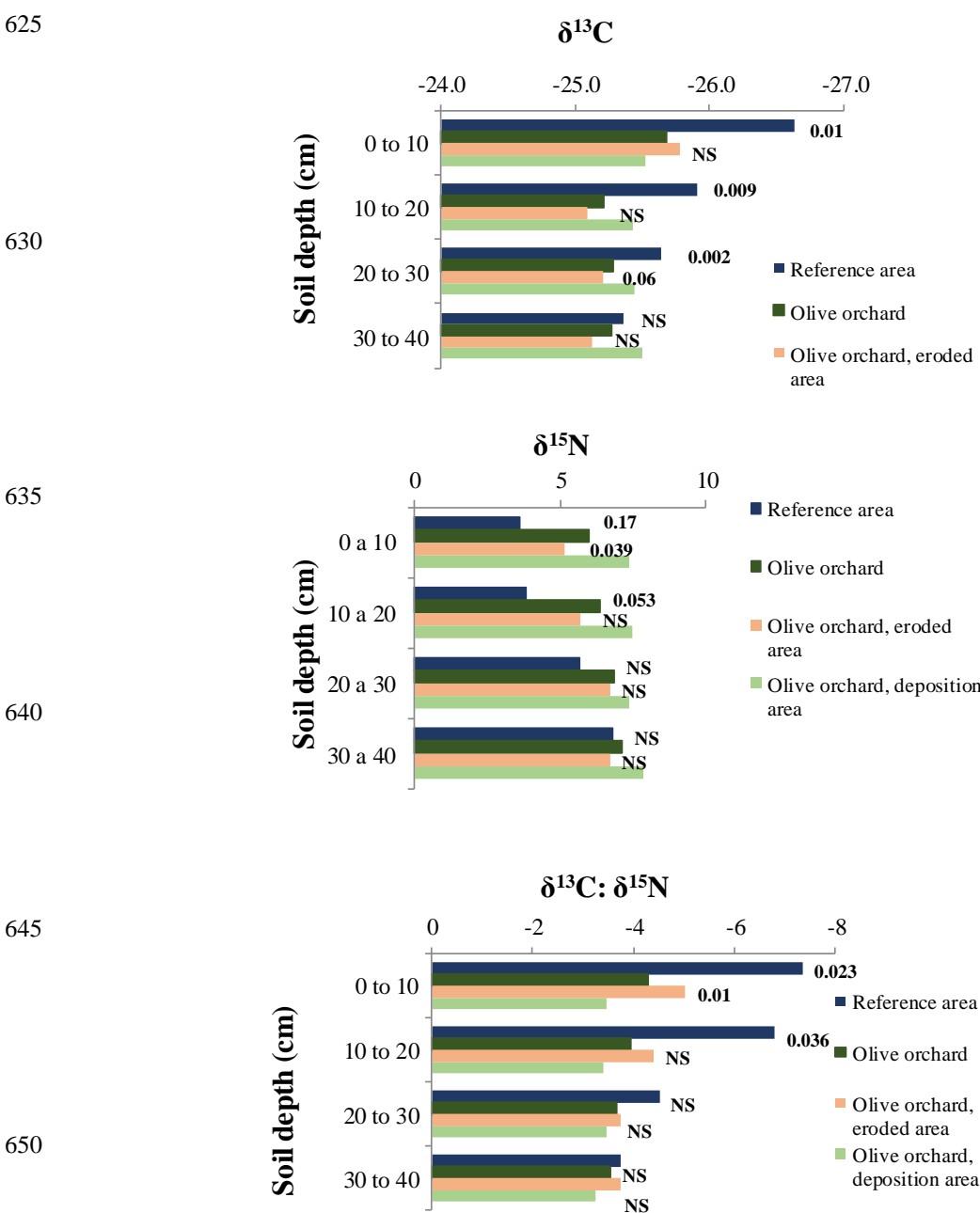

**Figure 9: Soil available phosphorus (P$_{avail}$), organic nitrogen (N$_{org}$), aggregate stability (W$_{sagg}$) and Soil Degradation Index (SDI)l by depth comparing eroded vs. deposition areas within the olive orchard. Labels in bars indicate the p-value according to a one-way ANOVA comparing areas for the same soil depth.**

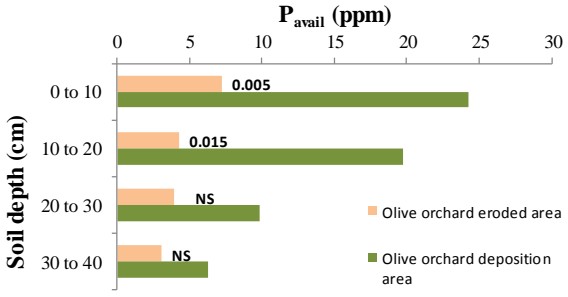
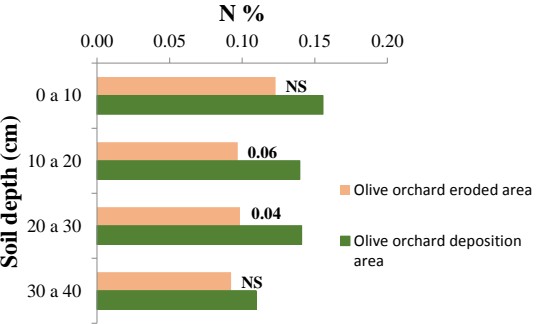

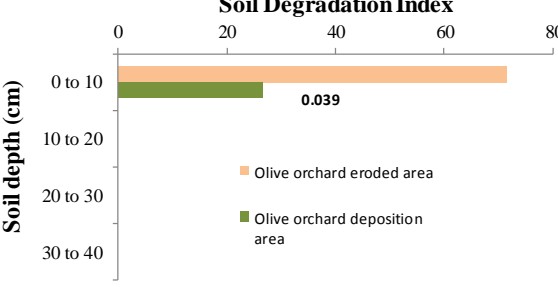
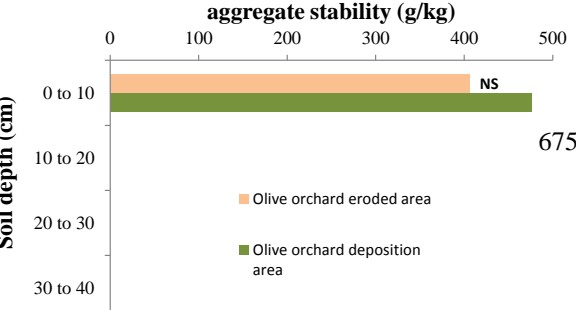

**Figure 10: Scores on principal components 1 and 2 (PC1, PC2) for sampling points in the eroded and deposition areas in the olive orchard.**

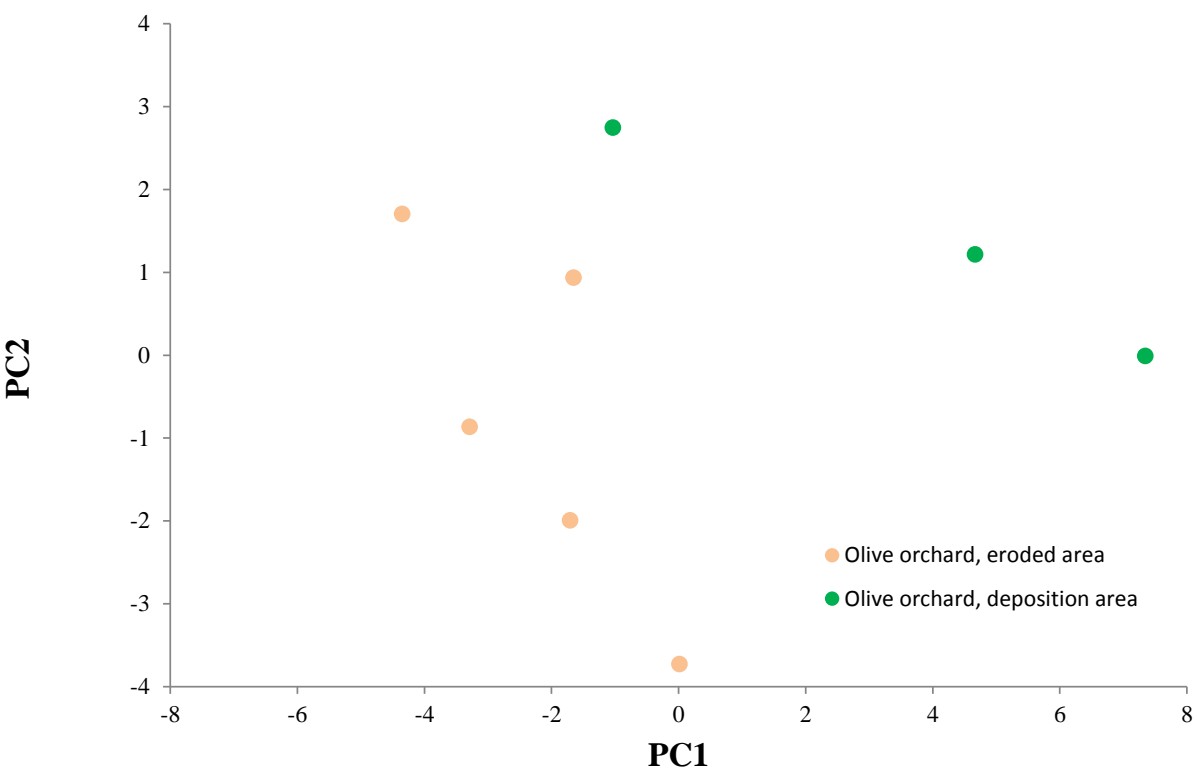

**Figure 11: Best correlations between some of the soil properties and erosion/deposition rates in the olive orchard.**

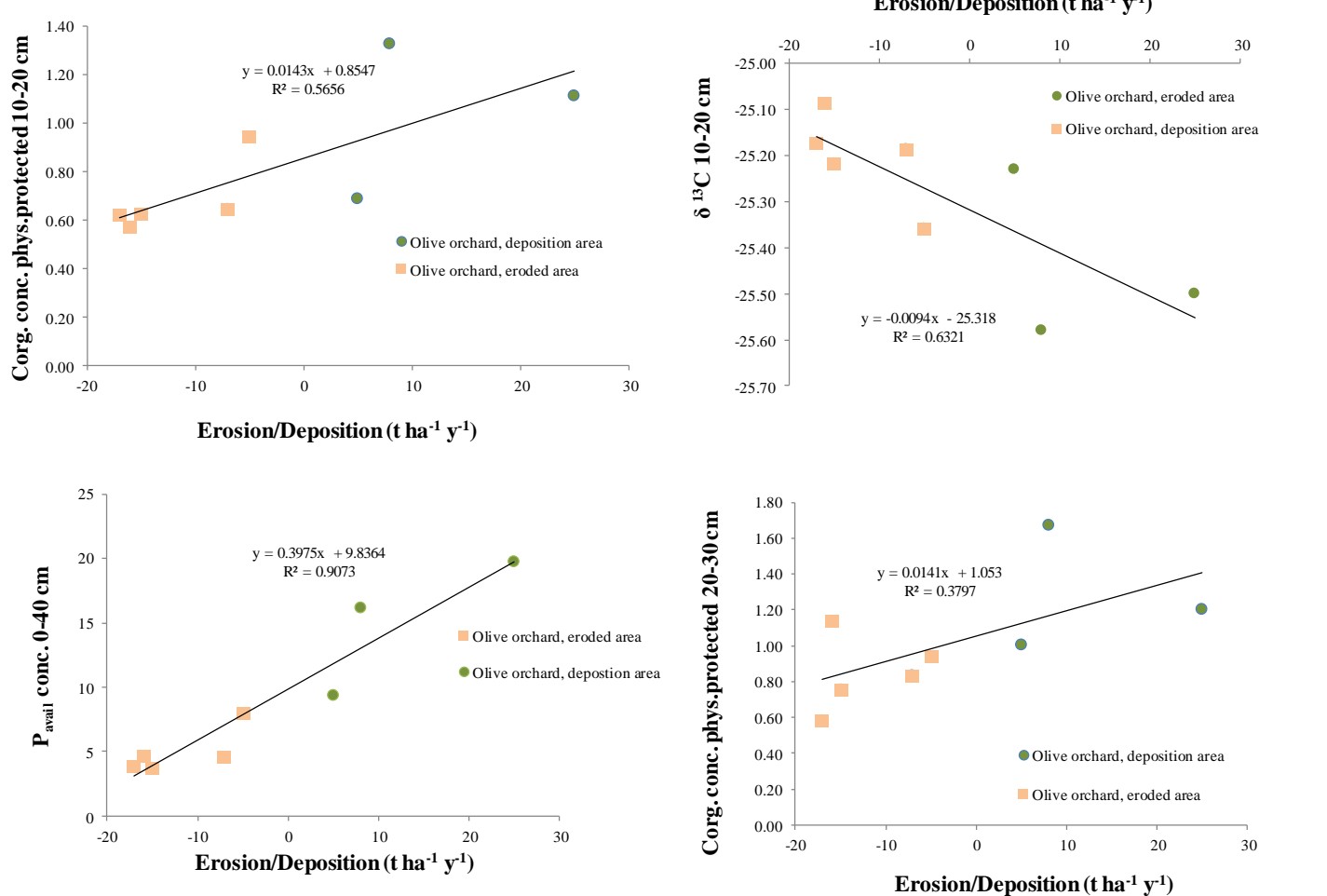

**Table 1: Location of the sampling points along the transect and associated soil redistribution rates derived from the** [137]Cs **technique (adapted from Mabit et al., 2012). Negative values indicate net erosion and positive values net deposition.**

| Point # | Code | Distance in transect (m) | Elevation (m) | Erosion/deposition rate (t ha$^{-1}$ yr$^{-1}$) |
|---------|------|--------------------------|---------------|---------------------------------|
| 1 | Cs1 | 0 | 1044 | -5.2 |
| 2 | Cs3 | 66.4 | 1032.8 | -17.8 |
| 3 | Cs5 | 125.0 | 1017.8 | -7.1 |
| 4 | Cs7 | 179.0 | 1006.8 | -16.3 |
| 5 | Cs12 | 312.2 | 986.8 | -15.2 |
| 6 | Cs13 | 338.1 | 984.8 | 5.9 |
| 7 | Cs15 | 388.8 | 981.8 | 24.7 |
| 8 | Cs17 | 429.5 | 979.8 | 8.8 |

**Table 2: Results of the two-way ANOVA analysis of soil organic carbon concentration, $C_{org}$ (%), in different fractions and in bulk soil. In A) area refers to reference site vs. olive orchard and in B) area refers to eroded vs. deposition areas in the olive orchard. NS stands for Not Significant.**

A)

| Model | Bulk soil | $C_{org}$ fraction | | | |
| --- | --- | --- | --- | --- | --- |
| | | Not protected | Physically protected | Chemically protected | Biochemically protected |
| Area (A) | 0.0000 | 0.0000 | 0.0000 | 0.0000 | 0.0000 |
| Depth (D) | 0.0023 | 0.0022 | 0.0061 | 0.0190 | NS |
| A x D | NS | NS | NS | NS | 0.0300 |

B)

| Model | Bulk soil | $C_{org}$ fraction | | | |
| --- | --- | --- | --- | --- | --- |
| | | Not protected | Physically protected | Chemically protected | Biochemically protected |
| Area (A) | 0.0198 | 0.0400 | 0.0077 | 0.0299 | NS |
| Depth (D) | 0.0081 | 0.0070 | 0.0058 | 0.0055 | 0.0847 |
| A x D | NS | NS | NS | NS | NS |

**Table 3: Results of the two-way ANOVA analysis of the distribution of the total soil organic carbon content in the soil among the different fractions of soil organic carbon, $C_{org}$. In A) area refers to reference site vs. olive orchard and in B) area refers to eroded vs. deposition areas in the olive orchard. NS stands for Not Significant.**

A)

| Model | $C_{org}$ fraction | | | |
|---|---|---|---|---|
| | Not protected | Physically protected | Chemically protected | Biochemically protected |
| Area (A) | 0.0000 | 0.0000 | 0.0000 | NS |
| Depth (D) | NS | NS | 0.0640 | NS |
| A x D | NS | 0.0059 | NS | NS |

B)

| Model | $C_{org}$ fraction | | | |
|---|---|---|---|---|
| | Not protected | Physically protected | Chemically protected | Biochemically protected |
| Area (A) | 0.091 | 0.0881 | NS | NS |
| Depth (D) | 0.051 | 0.0214 | NS | 0.033 |
| A x D | NS | NS | NS | NS |

**Table 4: Results of the two-way ANOVA analysis of the stable isotopes signal. In A) area refers to reference site vs. olive orchard and in B) area refers to eroded vs. deposition areas in the olive orchard. NS stands for Not Significant.**

**A)**

| Model | $\delta\,^{13}C$ | $\delta\,^{15}N$ | $\delta\,^{13}C : \delta\,^{15}N$ |
|---|---|---|---|
| Area (A) | 0.0001 | 0.002 | 0.002 |
| Depth (D) | 0.0026 | 0.0175 | 0.029 |
| A x D | NS | NS | NS |

**B)**

| Model | $\delta\,^{13}C$ | $\delta\,^{15}N$ | $\delta\,^{13}C : \delta\,^{15}N$ |
|---|---|---|---|
| Area (A) | NS | 0.01 | 0.01 |
| Depth (D) | NS | NS | NS |
| A x D | NS | NS | NS |

**Table 5: Results of the two-way ANOVA analysis of some soil physical and chemical properties comparing eroded vs. deposition areas in the olive orchard. NS stands for Not Significant.**

| Model | $N_{org}$ | $P_{avail}$ | Bulk density |
|---|---|---|---|
| Area (A) | 0.0000 | 0.01 | NS |
| Depth (D) | 0.0009 | NS | NS |
| A x D | NS | NS | NS |

**Table 6: Loads of selected variables in the PCA for the third first principal components (PC). Values in brackets below PC1, 2 and 3 indicate the percentage of variance explained by this PC. Variables in bold are those with a load higher than 90% of the variable with the maximum load for this PC. Conc. refers to Corg. concentration for this fraction, and Frac. means the relative contribution of this fraction to the total Corg for this soil depth.**

| Variable | PC1 (55.8) | PC2 (17.6) | PC3 (13.2) |
|---|---|---|---|
| **$P_{avail}$ 0-10 cm** | **0.2298** | 0.08765 | -0.07278 |
| **$P_{avail}$ 0_40 cm** | **0.2271** | 0.101 | -0.08455 |
| $\delta\ ^{13}C$ 0-10 cm | -0.03385 | 0.2861 | -0.3302 |
| $\delta\ ^{15}N$ 0-10 cm | 0.1941 | 0.1677 | 0.1836 |
| $N_{org}$ 0-10 cm | 0.2147 | -0.1375 | -0.0336 |
| $\delta\ ^{13}C{:}\delta\ ^{15}N$ 0-10 cm | 0.1594 | 0.249 | 0.2211 |
| $\delta\ ^{13}C$ 10-20 cm | **-0.2329** | 0.1461 | 0.04296 |
| $\delta\ ^{15}N$ 10-20 cm | 0.1856 | 0.07054 | 0.3121 |
| **$N_{org}$ 10-20 cm** | **0.2317** | -0.08699 | -0.1404 |
| $\delta\ ^{13}C{:}\delta\ ^{15}N$ 10-20 cm | 0.1637 | 0.08512 | 0.3309 |
| $\delta\ ^{13}C$ 20-30 cm | **-0.2316** | 0.06018 | 0.09789 |
| $\delta\ ^{15}N$ 20-30 cm | 0.1748 | -0.2656 | 0.1812 |
| $N_{org}$ 20-30 cm | 0.2176 | 0.09979 | -0.07458 |
| $\delta\ ^{13}C{:}\delta\ ^{15}N$ 20-30 cm | 0.1684 | -0.2982 | 0.1774 |
| **Corg conc. 10-20 cm** | **0.2421** | -0.02951 | -0.06407 |
| **Corg unpr. conc. 10-20 cm** | 0.2116 | 0.0331 | 0.0385 |
| **Corg unpr. Frac. 10-20 cm** | -0.09663 | -0.0509 | 0.364 |
| **Corg Phys. Pro. conc. 10-20 cm** | **0.2371** | -0.05728 | -0.1183 |
| **Corg Phys. Pro. Frac. 10-20 cm** | 0.123 | 0.163 | -0.02953 |
| **Corg Chem. Pro. conc. 10-20 cm** | 0.2072 | 0.03097 | -0.2335 |
| **Corg Chem. Pro. Frac. 10-20 cm** | -0.04095 | 0.09461 | **-0.4678** |
| **Corg conc. 20-30 cm** | **0.2326** | -0.1108 | -0.05737 |
| **Corg unpr. conc. 20-30 cm** | 0.2002 | -0.2372 | -0.05071 |
| **Corg unpr. Rel. 20-30 cm** | -0.132 | -0.3675 | -0.1379 |
| **Corg Phys. Pro. conc. 20-30 cm** | **0.2257** | 0.05677 | -0.01036 |
| **Corg Phys. Pro. Frac. 20-30 cm** | 0.09518 | 0.3528 | 0.1294 |
| **Corg Chem. Pro. conc. 20-30 cm** | **0.2323** | -0.06802 | -0.1366 |
| **Corg Chem. Pro. Frac. 20-30 cm** | 0.04565 | **0.4437** | -0.01278 |

**Table 7: Pearson correlation coefficients of lineal correlation between variables with highest loads on principal components (see Table 6).**

**\* means statistically significant at $\alpha<0.05$.**

| | Eros/dep. | $P_{avail}$ 0-10 cm | $P_{avail}$ 0-40 cm | $\delta^{13}C$ 10-20 cm | $N_{org}$ 10-20 cm | $\delta^{13}C$ 20-30 cm | Corg conc. 10-20 cm | Corg Phys. Pro. conc. 10-20 cm | Corg conc. 20-30 cm | Corg Phys. Pro. conc. 20-30 cm | Corg Chem. Pro. conc. 20-30 cm | Corg Chem. Pro. Frac. 20-30 cm |
|---|---|---|---|---|---|---|---|---|---|---|---|---|
| **Eros/dep.** | 1 | | | | | | | | | | | |
| **$P_{avail}$ 0-10 cm** | 0.746* | 1 | | | | | | | | | | |
| **$P_{avail}$ 0-40 cm** | **0.953*** | 0.857* | 1 | | | | | | | | | |
| **$\delta^{13}C$ 10-20 cm** | | -0.823* | -0.766* | 1 | | | | | | | | |
| **$N_{org}$ 10-20 cm** | 0.731* | 0.898* | 0.839* | -0.965* | 1 | | | | | | | |
| **$\delta^{13}C$ 20-30 cm** | **-0.792*** | -0.893* | -0.891* | 0.888* | -0.938* | 1 | | | | | | |
| **Corg conc. 10-20 cm** | 0.757* | 0.843* | 0.899* | -0.929* | 0.920* | -0.911* | 1 | | | | | |
| **Corg Phys. Pro. conc. 10-20 cm** | **0.755*** | 0.908* | 0.882* | -0.929 | 0.967* | -0.985* | 0.952* | 1 | | | | |
| **Corg conc. 20-30 cm** | | 0.786* | 0.799* | -0.965* | 0.927* | -0.846* | 0.9453* | 0.8942* | 1 | | | |
| **Corg Phys. Pro. conc. 20-30 cm** | | 0.837* | 0.756* | -0.846* | 0.817* | -0.714* | 0.8819* | 0.7928* | 0.901* | 1 | | |
| **Corg Chem. Pro. conc. 20-30 cm** | **0.742*** | 0.855* | 0.859* | -0.962* | 0.980* | -0.896* | 0.9515* | 0.9434* | 0.950* | 0.853* | 1 | |
| **Corg Chem. Pro. Frac. 20-30 cm** | | | | | | | | | | | | 1 |