# Peer review of "Variation of soil organic carbon, stable isotopes and soil quality indicators across an eroding-deposition catena in an historical Spanish olive orchard"

_SOIL, 2019_

## Referee Comment (RC1) · Anonymous Referee #1 · 29 Nov 2019

The manuscript is a case study in Spain to compare the variation of SOC, soil quantity indicators and isotopes induced by land-use change and erosion issues. The subjects addressed here were clear and worthy of investigation. Authors have chosen appropriate indexes (e.g. OC fractions, N, P et al.) to illustrate how olive orchard use coupled with soil erosion degrades soil quality, however, the data mining/ interpretation is insufficient and need to dig into further. In addition, the way of presenting results (Figures and Result section) are not well-structured and need to be reorganized.

Firstly, there are too many figures (29 figures) which are quite information poor. I highly recommend authors to reshape and combine some of them. For example, combine four individuals of Fig.3 into only one by a stacked bar chart (see attached Fig.1 as an example). Also, try to combine Fig. 2 A and B (Fig.2 as an example). Hopefully, it can reduce the number of figures from 29 to c. 11.

Secondly, a good dataset has been created in the manuscript but it is not deeply explored yet. Except for ANOVA, there are many statistics that would help out (e.g. PCA, correlation et al.). Why not try to correlate erosion/deposition rates with SOC or soil quality variables. In addition, authors have made ANOVA on reference vs orchard and orchard erosion vs orchard deposition, please give a further try to find a tendency on reference vs deposition if there are any.

Thirdly, please reorganize and give the subtitles for the Results section to make it clear and readable for audiences.

A couple of more comments:

1. L130 L170 How did you define unprotected, physically, chemically and biochemistry protected C? POM is unprotected C, iPOM physically protected C? Please clarify in Material & Method.

2. L120 Authors collected 13 micro pits from reference sites and 8 pits from olive orchard sites. Then you created one or three composite samples for fraction/isotopic measurement or measured all micro pigs as repeats?

3. L155 Please indicate the method you measuring bulk density, which was used in table 5

4. L205 Authors mentioned that "protected Corg in the reference and olive orchard area account for 87% and 64% of maximum soil stable Corg, respectively at the topsoil", it means reference area has a higher percentage of protected SOC than that of an olive orchard. This tendency is contrary to what has shown in Fig.5. How do you explain it? Please detail the way you calculated maximum soil stable Corg in Material & Method (insert equation for example?).

5. (L20 L300) authors suggested using $\delta$15N as a proxy to identify degraded areas; does annual input of 5 kg N-P fertilizers play a role in the dynamic of $\delta$15N?

Please also note the supplement to this comment:
https://www.soil-discuss.net/soil-2019-59/soil-2019-59-RC1-supplement.pdf

[Figure]

**Fig. 1.** example of re-shaping Fig. 3

[Figure]

corg. concenvration (%)

M    2.0    5.0

0-10 cm → Reference area deposits    eroded

10-20 cm    R    O

combining Fig. 2 A and B

**Fig. 2.** example of re-shaping Fig. 2

**Supplement:**

[revised manuscript text omitted]

---

## Referee Comment (RC2) · Anonymous Referee #2 · 3 Dec 2019

General comments

This paper focuses on the impact of long-term erosion and deposition processes on different soil parameters, especially bulk soil organic carbon and its fractions, within an historical olive orchard in Andalusia, Spain. The purpose of this study is worthy giving the importance of olive orchards and intense erosion processes in Mediterranean region. The soil parameters used to illustrate the impact of land use and erosion-deposition processes on soil quality have been well chosen. However, I have many concerns about the methodology, the data analysis, and the structuration of the

manuscript.

Different points of the 'materials and methods' section should be completed and more detailed as the sampling method and the method used to calculate the Corg stocks. Also, how the samples of the reference site were used in the data analysis is fuzzy to me. You'll find my related questions/requests in the specific comments below.

Whereas the authors chose well the parameters to study here and gather an interesting dataset, this latter seems insufficiently analyzed. I agree with referee #1, the authors should dig a bit further and try to better synthetized the results via fewer but more synthetic figures. Moreover, I have some serious concern about the way the Corg stocks and saturation capacity have been computed and treated.

The authors could review the 'results' and 'discussion' sections accordingly to new data analysis and figures. Please, could you better structures these sections and add sub-titles?

Please, find some specific comments and technical corrections below.

Specific comments

§2.1 'Description of the area'

As the study focuses on an erosion-deposition soil catena, an elevation map of the olive orchard or a topographic profile of the sampled transect locating the soil profiles could be appreciated.

§2.2 'Soil sampling'

The authors specified in the text that the reference site was sampled per 5 cm increments whereas the olive orchard was sampled per 10 cm increments. How did the authors compute values of soil parameters in reference site for the 10 cm increments?

All the results presented in the results section concerned the 40 first cm of soil. The reference site was sampled 'until bedrock was reached (i.e., 0-5, 5-10, 10-15, 15-20cm)

and when possible. . . '(l.119-120). Does it mean that the number of sample by 10cm increment in reference site is not constant? If the bedrock can be reached at 20cm within the reference site, what are the implications for the olive orchard especially in eroded areas? What are the implications on the rock fragment content in the samples and the computation of the Corg stocks?

Could you specify somewhere what are the final numbers of values analysed by 10cm increments in the reference site and in the olive orchard please?

The sampling was performed by a mechanical soil core. Was it a percussion drilling machine? Was there any soil deflection/compaction of the samples due to the mechanical drilling, i.e. was there any consequence on the depths of the soil increments?

The Corg stocks were calculated in the study. How exactly? Did you assess the soil bulk density based on the volume and mass of the soil increments? What about the rock fragments?

§2.3 'Physico-chemical analysis'

Corg concentration were determined according to Walkley and Black method. Did you apply a coefficient of correction to the raw data in order to take into account for the incomplete oxidation? This correction factor may vary from 1 to 1.6 depending on land use, soil texture, organic matter quality, sampling depth or climate. You compare two sites with different land uses, texture and organic matter quality (as highlighted by the fractionation results), and different depths.

You determined the theoretical values of stable carbon saturation based on the soil particle analysis. Could you specify exactly which model you used, with the values of the parameters, please? (See my comments below concerning the results section).

§3 'Results' l. 197-199: A more correct way to compare soil Corg stocks between different landuses is on equivalent soil mass.

l. 200-204: did you invert in the values of texture distribution between the reference

and olive orchard sites? If you have estimated the theoretical values of stable carbon saturation based on the content of particles <2$\mu$m (l. 205), the olive orchard should have a higher potential than the reference site according to the clay contents proposed here, i.e. 41 and 30% in the orchard and reference site respectively. Concerning the values of theoretical stable carbon saturation, could you precise the model used to compute them please? The values you proposed (i.e., 1.94 and 1.15%C; l.205) can't be achieved based on the model a proposed by Hassink & Whitmore (1997) in the Table 4.

§4 'Discussion'

l. 276: the value is 1.19 or 1.15%C as proposed line 205?

l. 278-280 : here, the authors affirmed that the land degradation reduced the soil capacity for Corg stabilization. If the authors well used the model fitted by Hassink and Whitmore in 1997 ('As proposed by Hassink and Whitmore (1997), theoretical values of carbon saturation were established from the soil particle analysis' l. 158-159), they know that basically the model is in the form : X = a * clay content + b with X the soil capacity for Corg stabilization, a and b some constants. As the soils in the reference and in the olive orchard have different clay content, they have different capacity for Corg stabilization! Here, it is like the authors were affirming that the land degradation has changed soil texture. . . I need more explanation and proof, please.

Technical corrections

Figure 1: Please, could you add bar scales or precise the olive orchard size in the part §2.1?

---

## Referee Comment (RC3) · Anonymous Referee #3 · 3 Dec 2019

soil-2019-59 this study examines changes in selected soil properties (SOC and SOC fractions, P available P and organic N) related to soil quality and explores the application of stable isotopes as indicators of soil degradation (13C and 15N) in an Calcic Cambisol under different land uses (open Mediterranean forest and orchard) in the southwestern region of Spain. Further, authors evaluated changes in the mentioned soil properties and water stable aggregates due to soil redistribution processes comparing eroded vs depositional sites within the olive orchard (areas previously identified by 137Cs technique). Please see below some comments: Line 23 deposition is nondegraded? Clarify Lines 22-25 I miss results concerning 13C Line 31 Although is a text extract with meaningful information. I suggest "which seeks to increase global soil organic matter stocks by 0.4 percent per year as a compensation for the global anthropogenic C emissions" Lines 33-34 split the paragraph into two sentences Line 41 This part seems disconnected from the previous one (soil degradation & soil quality). I suggest move this part to line 41 "Olive cultivation has been linked to severe environmental issues including the acceleration of erosion and soil degradation (e.g. Beaufoy, 2001, Scheidel and Krausmann, 2011). In fact, soil degradation is ... (Gómez, 2014)" Line 51 what is the reason for? Lines 58-59? Please rewrite to improve the readability of the text Line 85 It would be very illustrative to include the 137Cs reference value and sd Line 109 State exactly the plant species (shrubs and annual grasses) Line 120 Specify number of soil profiles deeper than 20 cm; excavation method is diddretn than mechanical method for soil sampling? Please include type of core sampler (automatic or manual soil core sampler) Line 125 A similar table for the two reference transects could be included (137Cs inventories since SRR are not applicable in ref site) Line 139 with sodium polytungstate Line 145 Explain in detail acid hydrolysis procedure: acid attack (acid concentration, time, temperature) and preparation for carbon analysis. Include a reference of the method. Line 163 Clarify the number of soil samples at similar soil depth and considered for statistical analysis Line 174 fractions Line 206 topsoil is 0-10 cm? Lines 212-216 This part should be extended and explained in depth Line 294 I consider there is no evidences from results for this statement (indicate selective deposition of soil aggregates). Please revise

---

## Author Comment (AC1) · 20 Dec 2019

We would like to thank to the Reviewer for the careful assessment of the manuscript and the helpful suggestions to improve the quality of our work.

We detail below each of the reviewer's comments and how we plan to address those suggestions in a revised version of the manuscript that we have not uploaded by the time of closing the period for posting comments due to the lack of time to prepare a convenient review. For the shake of clarity, the original comments by R1 are between

quotation marks.

"General comments: The manuscript is a case study in Spain to compare the variation of SOC, soil quantity indicators and isotopes induced by land-use change and erosion issues. The subjects addressed here were clear and worthy of investigation. Authors have chosen appropriate indexes (e.g. OC fractions, N, P et al.) to illustrate how olive orchard use coupled with soil erosion degrades soil quality, however, the data mining/ interpretation is insufficient and need to dig into further. In addition, the way of presenting results (Figures and Result section) are not well-structured and need to be reorganized." After reading the reviewer's comments we agree that the dataset deserves a more thorough analysis in the manuscript, part of which was done but was discarded (erroneously) trying to have a clearer manuscript. In addition, data presentation can be organized by combining several bar charts into one using bars of different colours and textures.

"Comment 1: Firstly, there are too many figures (29 figures) which are quite information poor. I highly recommend authors to reshape and combine some of them. For example, combine four individuals of Fig.3 into only one by a stacked bar chart (see attached Fig.1 as an example). Also, try to combine Fig. 2 A and B (Fig.2 as an example). Hopefully, it can reduce the number of figures from 29 to c. 11." We will combine Figure 2a and b into a single Figure 2 combining the 4 bars into one graph. For Figures 3a, b, c, d and Figures 4a, b, c, d, the four bar charts of each Figure will be merged into one distinguishing among treatments and depths using different colours and textures. For Figures 5a, b, c, d and Figure6a, b, c, d, the four bar charts in each one will be combined into one using a cumulative bar chart.

"Comment 2: Secondly, a good dataset has been created in the manuscript but it is not deeply explored yet. Except for ANOVA, there are many statistics that would help out (e.g. PCA, correlation et al.). Why not try to correlate erosion/deposition rates with SOC or soil quality variables. In addition, authors have made ANOVA on reference vs orchard and orchard erosion vs orchard deposition, please give a further try to find

a tendency on reference vs deposition if there are any." We will perform a correlation analysis between the normalized evaluated soil parameters vs. erosion/deposition rates and slope length for the topsoil values in the orchard area. Additionally, we will perform an exploratory analysis using PCA using the evaluated soil properties comparing the three areas (olive eroded, olive deposition, reference area). "Comment 3: Thirdly, please reorganize and give the subtitles for the Results section to make it clear and readable for audiences." We will reorganize the results and discussion sections using clear subtitle names to facilitate a more clear reading and understanding.

" Additional minor comments" "1. L130 L170 How did you define unprotected, physically, chemically and biochemistry protected C? POM is unprotected C, iPOM physically protected C? Please clarify in Material & Method." We agree with the reviewer comment. In the revision version of the manuscript we have added the following sentences: This three-step process isolates a total of 12 fractions and it is based on the assumed link between the isolated fractions and the protection mechanisms involved in the stabilization of organic C (Six et al., 2002). The unprotected pool includes the POM and LF fractions, isolated in the first and second fractionation steps, respectively. The physically protected SOC consists of the SOC measured in the microaggregates. It includes not only the iPOM but also the hydrolysable and non-hydrolysable SOC of the intermediate fraction (53–250 $\mu$m). The chemically and biochemically protected pools correspond to that hydrolysable and non-hydrolysable SOC in the fine fraction (< 53 $\mu$m), respectively.

"2. L120 Authors collected 13 micro pits from reference sites and 8 pits from olive orchard sites. Then you created one or three composite samples for fraction/isotopic measurement or measured all micro pigs as repeats?" This comment helps us to realized that there are unclear sections in the manuscript that need to be clarified. In the olive orchard area (8 points for core samples) we treated each point and depth as a single unit for all the analysis (fraction, isotopic, . . .). In the reference area we sampled 13 pits and all of them were used for the isotopic analysis of 137Cs shown in Table 1,

while only 4 of them were used to determine the carbon fractions, and $\delta$ 15N and $\delta$ 13C isotopic analysis.

"3. L155 Please indicate the method you measuring bulk density, which was used in table 5." Soil bulk density in Table 5 was measured using a hand cylindrical core sampler with a volume of 100 cm3.

"4. L205 Authors mentioned that "protected Corg in the reference and olive orchard area account for 87% and 64% of maximum soil stable Corg, respectively at the top-soil", it means reference area has a higher percentage of protected SOC than that of an olive orchard. This tendency is contrary to what has shown in Fig.5. How do you explain it? Please detail the way you calculated maximum soil stable Corg in Material & Method (insert equation for example?)". 1.- As mentioned in line 204, maximum capacity to stabilised SOC in the reference and olive orchard sites was estimated according to Hassink and Whitmore (1997). The amount of protected C in the reference and olive orchard soils accounted for 87 % and 64 % of the maximum capacity, respectively. In the revised version of the manuscript these percentages will change as reviewer 2 found an error in our calculations that has been corrected. Nevertheless, in the new recalculations, the amount of protected C respect to maximum soil stable Corg in reference site doubled that of the olive groves soils.

2.- Figure 5 shows the percentages of total organic carbon in each of the fraction. The fact that the percentage of SOC in protected fraction in olive grove soils is higher than that of the reference soil is due to SOC concentration in the reference site which is much higher than that of the olive grove soil, although in the former most of the SOC is unprotected, therefore the contribution of the protected SOC respect to the total in the reference soil could be lower than that of the olive grove soil, even when protected SOC in the reference site is higher than that of the olive grove site.

"5. (L20 L300) authors suggested using $\delta$15N as a proxy to identify degraded areas; does annual input of 5 kg N-P fertilizers play a role in the dynamic of $\delta$15N?" We

agree that in the revised version of the manuscript the influence of the NP fertilizer in modifying the $\delta$15N in relation to the reference area, probably with an slight enrichment see Alison et al. (2007), need to be considered too. Alison S. Bateman & Simon D. Kelly (2007) Fertilizer nitrogen isotope signatures, Isotopes in Environmental and Health Studies, 43:3, 237-247, DOI: 10.1080/10256010701550732

"6. Please also note the supplement to this comment: https://www.soil-discuss.net/soil-2019-59/soil-2019-59-RC1-supplement.pdf" We have checked the comments made in the annotated version of the manuscript and these indications will be incorporated in the revised version of the manuscript.

---

## Author Comment (AC2) · 20 Dec 2019

We appreciate the contribution of the reviewer to improve the quality of our work through a careful assessment of the manuscript and helpful suggestions.

We detail below each of the reviewer's comments and how we plan to address those suggestions in a revised version of the manuscript that we have not uploaded by the time of closing the period for posting comments due to the lack of time to prepare a convenient review. For the shake of clarity, the original comments by R2 are between

quotation marks.

"General comments: This paper focuses on the impact of long-term erosion and deposition processes on different soil parameters, especially bulk soil organic carbon and its fractions, within an historical olive orchard in Andalusia, Spain. The purpose of this study is worthy giving the importance of olive orchards and intense erosion processes in Mediterranean region. The soil parameters used to illustrate the impact of land use and erosion-deposition processes on soil quality have been well chosen. However, I have many concerns about the methodology, the data analysis, and the structuration of the manuscript. Different points of the 'materials and methods' section should be completed and more detailed as the sampling method and the method used to calculate the Corg stocks. Also, how the samples of the reference site were used in the data analysis is fuzzy to me. You'll find my related questions/requests in the specific comments below. Whereas the authors chose well the parameters to study here and gather an interesting dataset, this latter seems insufficiently analyzed. I agree with referee #1, the authors should dig a bit further and try to better synthetized the results via fewer but more synthetic figures. Moreover, I have some serious concern about the way the Corg stocks and saturation capacity have been computed and treated. The authors could review the 'results' and 'discussion' sections accordingly to new data analysis and figures. Please, could you better structures these sections and add sub-titles? Please, find some specific comments and technical corrections below." We will expand the details on methodology and sampling, e.g. giving more details on the sampling equipment, calculation of the Corg stocks, etc., and improve the clarity of the data analysis following the recommendations of the three reviewers. Additionally, we will re-check the calculations on the Corg stocks and saturation capacity and structure the results and discussion sections also including subtitles to facilitate the reading.

"Comment §2.1 'Description of the area': As the study focuses on an erosion-deposition soil catena, an elevation map of the olive orchard or a topographic profile of the sampled transect locating the soil profiles could be appreciated." We will provide in

the supplementary material a transect showing the elevation of the sampled areas and an elevation map including the orchard and the reference area.

"Comment §2.2 'Soil sampling': The authors specified in the text that the reference site was sampled per 5 cm increments whereas the olive orchard was sampled per 10 cm increments. How did the authors compute values of soil parameters in reference site for the 10 cm increments? All the results presented in the results section concerned the 40 first cm of soil. The reference site was sampled 'until bedrock was reached (i.e., 0-5, 5-10, 10-15, 15-20cm). '(l.119-120). Does it mean that the number of sample by 10cm increment in reference site is not constant? If the bedrock can be reached at 20cm within the reference site, what are the implications for the olive orchard especially in eroded areas? What are the implications on the rock fragment content in the samples and the computation of the Corg stocks?" The sampling in the reference area was done manually at 5 cm depth intervals (e.g. 0-5 and 5-10 cm) and the samples were integrated to perform the analysis at 10 cm intervals (e.g. integrating 0-5 and 5-10 cm into one for 0-10 cm). We found a mistake in lines 119-120; the reference area was sampled until reaching bedrock which was located at least at 40 cm in all cases. The carbon, $\delta$13C and $\delta$15N analysis comes from samples from four of these pits, while the 137Cs analysis comes from the 13 pits. In all these pits bedrock was below 40 cm depth. Carbon stock calculations were made for the fine soil fraction (< 2 mm) after discounting rock or stone fragments larger than 2 mm and considering soil bulk density measured using the hand cylindrical core sampler with a volume of 100 cm3.

"Comment 3: Could you specify somewhere what are the final numbers of values analysed by 10cm increments in the reference site and in the olive orchard please?" Yes. In the revised version of the text we will include the information requested by the reviewer. The number of soil samples for each 10 cm increments was 4 and 8 from the reference area and olive orchard, respectively.

"Comment 4: The sampling was performed by a mechanical soil core. Was it a percussion drilling machine? Was there any soil deflection/compaction of the samples due to

the mechanical drilling, i.e. was there any consequence on the depths of the soil increments?" The mechanical soil sampling was made with a hydraulic core sample which gently rotates and push the core, and with the soil at a moisture content between 40 to 80 WHC, and therefore we did no need to hard drilling the soil, minimizing the compression of the soil samples. The sampling was made properly, insuring that the whole sample was taken for each given depth, abandoning the point and starting a new one if some problem arose (like a sample being only partially taken). A better explanation of this and the model of core equipment used will be included in a revised version of the manuscript.

"Comment 5: The Corg stocks were calculated in the study. How exactly? Did you assess the soil bulk density based on the volume and mass of the soil increments? What about the rock fragments¿' Soil carbon stock were calculated for the fine soil fraction after discounting rock or stone fragments larger than 2 mm, and considering bulk density which was measured following using the hand cylindrical core sampler with a volume of 100 cm3

"Comment §2.3 'Physico-chemical analysis': Corg concentration were determined according to Walkley and Black method. Did you apply a coefficient of correction to the raw data in order to take into account for the incomplete oxidation? This correction factor may vary from 1 to 1.6 depending on land use, soil texture, organic matter quality, sampling depth or climate. You compare two sites with different land uses, texture and organic matter quality (as highlighted by the fractionation results), and different depths." We thanks to the reviewers for this comment, as we have detected a mistake in the reference we used regarding the method for SOC determination. In all cases, SOC fractions and in the bulk soil, organic carbon concentrations were determined by using the wet oxidation sulfuric acid and potassium dichromate method of Anderson and Ingram (1993). We have corrected this in the revised version of the manuscript.

"Comment: You determined the theoretical values of stable carbon saturation based on the soil particle analysis. Could you specify exactly which model you used, with the

values of the parameters, please? (See my comments below concerning the results section)." See answer to concerns in result section.

"Comment §3 'Results': l. 197-199: A more correct way to compare soil Corg stocks between different land uses is on equivalent soil mass." We will compare the reference area and the olive orchard in equivalent soil mass following the procedure describe in Wend and Hauser, 2013. An equivalent soil mass procedure for monitoring soil organic carbon in multiple soil layers. European Journal of Soil Science doi: 10.1111/ejss.12002 , 2013

"Comment. 200-204: did you invert in the values of texture distribution between the reference and olive orchard sites? If you have estimated the theoretical values of stable carbon saturation based on the content of particles <2$\mu$m (l. 205), the olive orchard should have a higher potential than the reference site according to the clay contents proposed here, i.e. 41 and 30% in the orchard and reference site respectively. Concerning the values of theoretical stable carbon saturation, could you precise the model used to compute them please? The values you proposed (i.e., 1.94 and 1.15%C; l.205) can't be achieved based on the model a proposed by Hassink & Whitmore (1997) in the Table 4." We thank the reviewer for his comments on this topic as we have detected errors on our calculations regarding the theoretical values of stable carbon saturation. As reviewer has detected, the olive orchard soils have a higher potential than reference site. We have applied the model of Hassink & Whitmore (1997). According to this model, the theoretical value of protected SOC (g C kg soil-1) = 21,1 + (clay content (g kg-1 soil) x 0,0375. Considering that there were not significant differences in the soil clay content along the catena in the olive grove soils, the average theoretical protected SOC (%) is 3,63ïĆś0,19, where in the reference site averaged 3,24ïĆś0,11. According to these new values, protected soil Corg in the reference site and orchard soils accounted for 20.5ïĆś5.2 % and 49.8ïĆś11.5 % of the maximum soil stable Corg, respectively at the topsoil. After this amended, conclusions have not changed respect to the first submitted version of the manuscript. In the revised version of the manuscript

we have corrected the data: Texture distribution of the topsoil (0–10 cm) along the catena in the olive orchard presents an average clay, silt and sand content of 41, 37 and 22% and low variability, respectively (average coefficient of variation of 17%) without significant changes between the erosion and deposition areas. In the reference area, the soil has an average clay, silt and sand content of 30, 31 and 39% respectively, also with a homogeneous distribution across the sampling area (coefficient of variation of 10%). According to the Hassink and Whitmore (1997) model, the percentages of organic carbon of maximum soil stable Corg are of 3.63ïĆś0.19 and 3.24ïĆś0.11 % in the reference site and olive orchard, respectively. So, protected Corg in the reference and olive orchard areas account for 20.5ïĆś5.2 % and 49.8ïĆś11.5 % of the maximum soil stable Corg, respectively at the topsoil.

"Comment §4 'Discussion': l. 276: the value is 1.19 or 1.15%C as proposed line 205?" Reviewer is right in his/her concern on this issue, and we are deeply sorry on our errors on the calculations. The correct value is (3.64ïĆś0.23 %) and we have amended in the revised version of the manuscript: In fact, the protected Corg concentration in the topsoil of the olive orchard in the eroded area is about the 18.6ïĆś3.9 % of the upper limit of protected Corg (3.64ïĆś0.23 %) according to the model of Hassink and Whitmore (1997)

"Comment l. 278-280: here, the authors affirmed that the land degradation reduced the soil capacity for Corg stabilization. If the authors well used the model fitted by Hassink and Whitmore in 1997 ('As proposed by Hassink and Whitmore (1997), theoretical values of carbon saturation were established from the soil particle analysis' l. 158-159), they know that basically the model is in the form : X = a * clay content + b with X the soil capacity for Corg stabilization, a and b some constants. As the soils in the reference and in the olive orchard have different clay content, they have different capacity for Corg stabilization! Here, it is like the authors were affirming that the land degradation has changed soil texture... I need more explanation and proof, please." Reviewer is right. We will delete this sentence in the revised version of the manuscript.

"Technical corrections Figure 1: Please, could you add bar scales or precise the olive orchard size in the part §2.1?" Yes we will include a scale bar for this Figure.

---

## Author Comment (AC3) · 20 Dec 2019

We would like to express our appreciation to the Reviewer for the careful assessment of the manuscript and the helpful suggestions, which has help us to improve our work.

We detail below each of the reviewer's comments and how we plan to address those suggestions in a revised version of the manuscript that we have not uploaded by the time of closing the period for posting comments due to the lack of time to prepare a convenient review. For the shake of clarity, the original comments by R3 are between

quotation marks.

"General comment: This study examines changes in selected soil properties (SOC and SOC fractions, P available P and organic N) related to soil quality and explores the application of stable isotopes as indicators of soil degradation (13C and 15N) in an Calcic Cambisol under different land uses (open Mediterranean forest and orchard) in the southwestern region of Spain. Further, authors evaluated changes in the mentioned soil properties and water stable aggregates due to soil redistribution processes comparing eroded vs depositional sites within the olive orchard (areas previously identified by 137Cs technique). Please see below some comments: "

"Comment 1: Line 23 deposition is non degraded?" Yes, that is our hypothesis. We will clarify this in a revised version of the manuscript.

"Comment 2: Clarify Lines 22-25 I miss results concerning 13C" We will add one line concerning $\delta$ 13C results in the revised version of the manuscript.

"Comment 3, Line 31 Although is a text extract with meaningful information. I suggest "which seeks to increase global soil organic matter stocks by 0.4 percent per year as a compensation for the global anthropogenic C emissions" Lines 33-34 split the paragraph into two sentences." Yes we will edit and split this section into two sentences in a revised version of the manuscript.

"Comment 4: Line 41 This part seems disconnected from the previous one (soil degradation & soil quality). I suggest move this part to line 41 "Olive cultivation has been linked to severe environmental issues including the acceleration of erosion and soil degradation (e.g. Beaufoy, 2001, Scheidel and Krausmann, 2011). In fact, soil degradation is ... (Gómez, 2014)." We agree with the comment. We will edit this section in this way which looks more straightforward.

"Comment 5: Line 51 what is the reason for?" It also combined cultivation in very steep slopes and areas of high rainfall erosivity. We will edit this sentence to include this

evaluation in the revised version of the manuscript.

"Comment 6: Lines 58-59? Please rewrite to improve the readability of the text Line 85 It would be very illustrative to include the 137Cs reference value and sd Line 109 State exactly the plant species (shrubs and annual grasses)." We will include the 137Cs reference value and include a list of the most common shrubs and annual grasses in the study site.

"Comment 7 Line 120 Specify number of soil profiles deeper than 20 cm; excavation method is diddretn than mechanical method for soil sampling? Please include type of core sampler (automatic or manual soil core sampler)." We will revise the manuscript to clarify the sampling method and the number of soil profiles. In the reference area the sampling was made through manual excavation while in the olive orchard the sampling at 10 cm interval was performed using a hydraulic core sample which gently rotates and push the core. Soil moisture content was the adequate to avoid hard drilling the soil. This minimizes the compression of the samples. The sampling was made checking that the whole sample was taken for each given depth abandoning the point and starting a new one if some problem arose (like a sample being only partially taken). The bulk density values shown in Table 5 were obtained using the hand cylindrical core sampler with a volume of 100cm3. Regarding the number of samples and depths, there is a mistake in lines 119-120. The reference area was sampled until reaching bedrock which in same case was above 60 cm. In the case of the pits used for the carbon and isotopic analysis, 4 out of 13 were used, while all the 13 pits were used for the Cs137 analysis. In all cases these 13 pits reached 40 cm depth. Therefore, for the carbon and isotopic N and C analysis there were, for each soil depth (0-10, 10-20, 20-30, 30-40 cm) four replications in the reference area and 8 replications in the olive orchard.

"Comment 8: Line 125 A similar table for the two reference transects could be included (137Cs inventories since SRR are not applicable in ref site)." We will add a Table in the supplementary material indicating the elevation of the transects (which were in a flat area) and their 137Cs inventory.

"Comment 9: Line 139 with sodium polytungstate " This misprint will be corrected in the revised version of the manuscript.

"Comment 10: Line 145 Explain in detail acid hydrolysis procedure: acid attack (acid concentration, time, temperature) and preparation for carbon analysis. Include a reference of the method." As we mentioned in the material and method section (Physicochemical analysis) we have applied the method of Six et al. (2002) and modified by Stewart et al. (2009). Acid hydrolysis, described by Plante et al. (2006) consisted of incubating the samples (The silt+clay-size fraction from both the density flotation of the $53 - 250$ $\mu$m fraction and the initial dispersion and physical fractionation of the $< 53$ $\mu$m fraction) at 95 oC for 16 h in 25 ml of 6 M HCl. After hydrolysis, the suspension was filtered and washed with deionized water over a glass-fiber filter. Residues were dried at 60 oC and weighed. These fractions represent the non-hydrolyzable C fractions. The hydrolysable C fractions were determined by difference between the total organic C content of the fractions and the C contents of the non- hydrolyzable fractions. We have added this information in material and method section.

"Comment 11; Line 163 Clarify the number of soil samples at similar soil depth and considered for statistical analysis Line 174 fractions Line 206 topsoil is 0-10 cm?" It will be clarified in the text, clearly stating that for the carbon and isotopic N and C analysis there were, for each soil depth (0-10, 10-20, 20-30, 30-40 cm) four replications in the reference area and 8 replications in the olive orchard. In line 206 top-soil means 0-10 cm, this will also be clarified.

"Comment 12: Lines 212-216 This part should be extended and explained in depth." We will expand this section to provide an explanation.

"Comment 13: Line 294 I consider there is no evidences from results for this statement (indicate selective deposition of soil aggregates). Please revise". We will edit this section to indicate that this still as a plausible hypothesis rather than a fact demonstrated by our results.

---

## Author Response (AR2)

We would like to thank to the reviewers for the careful assessment of the manuscript and the helpful suggestions to improve the quality of our work.

We detail below each of the reviewer's comments and how we plan to address those suggestions in a revised version of the manuscript that we have not uploaded by the time of closing the period for posting comments due to the lack of time to prepare a convenient review. For the shake of clarity, the original comments by each reviewer are between quotation marks.

**Reviewer 1**

"General comments: The manuscript is a case study in Spain to compare the variation of SOC, soil quantity indicators and isotopes induced by land-use change and erosion issues. The subjects addressed here were clear and worthy of investigation. Authors have chosen appropriate indexes (e.g. OC fractions, N, P et al.) to illustrate how olive orchard use coupled with soil erosion degrades soil quality, however, the data mining/ interpretation is insufficient and need to dig into further. In addition, the way of presenting results (Figures and Result section) are not well-structured and need to be reorganized." After reading the reviewer's comments we agree that the dataset deserves a more thorough analysis in the manuscript, part of which was done but was discarded (erroneously) trying to have a clearer manuscript. In addition, data presentation can be organized by combining several bar charts into one using bars of different colours and textures.

"Comment 1: Firstly, there are too many figures (29 figures) which are quite information poor. I highly recommend authors to reshape and combine some of them. For example, combine four individuals of Fig.3 into only one by a stacked bar chart (see attached Fig.1 as an example). Also, try to combine Fig. 2 A and B (Fig.2 as an example). Hopefully, it can reduce the number of figures from 29 to c. 11."

We have combined Figure 2a and b into a single Figure 2 combining the 4 bars into one graph. For Figures 3a, b, c, d and Figures 4a, b, c, d, the four bar charts of each Figure have been merged into one distinguishing among treatments and depths using different colours and textures. For Figures 5a, b, c, d and Figure6a, b, c, d, the four bar charts in each one have been combined into one using a cumulative bar chart.

"Comment 2: Secondly, a good dataset has been created in the manuscript but it is not deeply explored yet. Except for ANOVA, there are many statistics that would help out (e.g. PCA, correlation et al.). Why not try to correlate erosion/deposition rates with SOC or soil quality variables. In addition, authors have made ANOVA on reference vs orchard and orchard erosion vs orchard deposition, please give a further try to find a tendency on reference vs deposition if there are any."

Additionally, have expanded this analysis performing an exploratory analysis using PCA using the evaluated soil properties comparing the two areas within the olive orchard (eroded vs. deposition), complemented with a correlation analysis between the variables identified in this exploratory analysis and erosion/deposition rates in the orchard area.

"Comment 3: Thirdly, please reorganize and give the subtitles for the Results section to make it clear and readable for audiences."

We have reorganized the results and discussion sections using clear subtitle names to facilitate a more clear reading and understanding.

**" Additional minor comments"**

"1. L130 L170 How did you define unprotected, physically, chemically and biochemistry protected C? POM is unprotected C, iPOM physically protected C? Please clarify in Material & Method."

We agree with the reviewer comment. In the revision version of the manuscript we have clarified the definition of SOC fraction. We have added the following sentences: (lines 139-141) This three-step process isolates a total of 12 fractions and it is based on the assumed link between the isolated fractions and the protection mechanisms involved in the stabilization of organic C (Six et al., 2002) (lines 155-159) The unprotected pool includes the POM and LF fractions, isolated in the first and second fractionation steps, respectively. The physically protected SOC consists of the SOC measured in the microaggregates. It includes not only the iPOM but also the hydrolysable and non-hydrolysable SOC of the intermediate fraction (53–250  $\mu$ m). The chemically and biochemically protected pools correspond to that hydrolysable and non-hydrolysable SOC in the fine fraction (< 53  $\mu$ m), respectively. In addition, we have added how organic carbon concentration was measured in the SOC fractions: (lines 159-161).

"2.L120 Authors collected 13 micro pits from reference sites and 8 pits from olive orchard sites. Then you created one or three composite samples for fraction/isotopic measurement or measured all micro pigs as repeats?"

We thanks to the reviewer for this comment, which has helped us to realize that there are unclear sections in the manuscript that need to be clarified. In the olive orchard area (8 points for core samples) we treated each point and depth as a single unit for all the analysis (fraction, isotopic, ...). In the reference area we sampled 13 pits and all of them were used for the isotopic analysis of 137Cs, as it is shown in Table 1, while only 4 of them were used to determine the carbon fractions, and  $\delta$  15N and  $\delta$  13C isotopic analysis. This has been made clearer in the material and method section, lines 121-133.

**"3. L155 Please indicate the method you measuring bulk density, which was used in table 5."**

Soil bulk density in Table 5 was measured using a hand cylindrical core sampler with a volume of 100 cm3 (lines 175-176 of the revised version of the manuscript).

**"4.** L205 Authors mentioned that "protected Corg in the reference and olive orchard area account for 87% and 64% of maximum soil stable Corg, respectively at the topsoil", it means reference area has a higher percentage of protected SOC than that of an olive orchard. This tendency is contrary to what has shown in Fig.5. How do you explain it? Please detail the way

you calculated maximum soil stable Corg in Material & Method (insert equation for example?)".

1.- As mentioned in line 228, maximum capacity to stabilised SOC in the reference and olive orchard sites was estimated according to Hassink and Whitmore (1997). According to this model, the theoretical value of protected SOC (g C kg soil-1) is calculated as = 21,1 + (clay content (g kg-1 soil) x 0,0375. Considering that there were not significant differences in the soil clay content along the catena in the olive grove soils, the average theoretical protected SOC (%) is  $3,63\pm0,19$ , whereas in the reference site averaged  $3,24\pm0,11$ . According to these new values, protected soil Corg in the reference site and orchard soils accounted for  $49.8\pm11.5$  % and  $20.5\pm5.2$  % of the maximum soil stable Corg, respectively at the topsoil. This has been noted in lines of the revised manuscript 303-305.

Figure 5 shows the contribution (in %) of the different fraction to the total soil organic carbon. Most (about 78 %) of the total SOC in the olive groves soils is protected, however because total SOC is relatively low in these soils (about 0.9 %), the concentration of protected carbon is relatively low (about 0.7 %), especially compared to the maximum theoretical value of protected SOC (3.63 %). In the reference site, the contribution of the protected SOC respect to the total SOC is much lower (about 35 %, as most of the soil organic carbon in unprotected), but because total SOC concentration is high (about 4.9 %), the concentration of protected carbon is higher (about 1.7 %) than that of the olive grove soils, and much closer to the maximum theoretical value of protected SOC (3.24 %) than the olive grove soils.

"5. (L20 L300) authors suggested using  $\delta$ 15N as a proxy to identify degraded areas; does annual input of 5 kg N-P fertilizers play a role in the dynamic of  $\delta$ 15N?"

We agree, and in the revised version of the manuscript the influence of the NP fertilizer in modifying the  $\delta$ 15N in relation to the reference area, probably with an slight enrichment see Alison et al. (2007), need to be considered too, lines 324-327 of the revised version of the manuscript.

Alison S. Bateman & Simon D. Kelly (2007) Fertilizer nitrogen isotope signatures, Isotopes in Environmental and Health Studies, 43:3, 237-247, DOI: 10.1080/10256010701550732

**"6. Please also note the supplement to this comment: https://www.soil-discuss.net/soil-2019- 59/soil-2019-59-RC1-supplement.pdf"**

We have checked the comments made in the annotated version of the manuscript and these indications will be incorporated in the revised version of the manuscript.

**Reviewer 2**

"General comments: This paper focuses on the impact of long-term erosion and deposition processes on different soil parameters, especially bulk soil organic carbon and its fractions, within an historical olive orchard in Andalusia, Spain. The purpose of this study is worthy giving the importance of olive orchards and intense erosion processes in Mediterranean region. The soil parameters used to illustrate the impact of land use and erosion-deposition processes on soil quality have been well chosen. However, I have many concerns about the methodology, the data analysis, and the structuration of the manuscript. Different points of the 'materials and methods' section should be completed and more detailed as the sampling method and the method used to calculate the Corg stocks. Also, how the samples of the reference site were used in the data analysis is fuzzy to me. You'll find my related questions/requests in the specific comments below. Whereas the authors chose well the parameters to study here and gather an interesting dataset, this latter seems insufficiently analyzed. I agree with referee #1, the authors should dig a bit further and try to better synthetized the results via fewer but more synthetic figures. Moreover, I have some serious concern about the way the Corg stocks and saturation capacity have been computed and treated. The authors could review the 'results' and 'discussion' sections accordingly to new data analysis and figures. Please, could you better structures these sections and add sub-titles? Please, find some specific comments and technical corrections below."

We have expanded the details on methodology and sampling, e.g. giving more details on the sampling equipment, calculation of the Corg stocks, etc., and improve the clarity of the data analysis following the recommendations of the three reviewers. Additionally, we have rechecked the calculations on the Corg stocks and saturation capacity and structured the results and discussion sections also including subtitles to facilitate the reading.

"Comment §2.1 'Description of the area': As the study focuses on an erosion-deposition soil catena, an elevation map of the olive orchard or a topographic profile of the sampled transect locating the soil profiles could be appreciated."

We have provide in the revised Figure 1 a transect showing the elevation of the sampled areas and an elevation map including the orchard and the reference area.

"Comment §2.2 'Soil sampling': The authors specified in the text that the reference site was sampled per 5 cm increments whereas the olive orchard was sampled per 10 cm increments. How did the authors compute values of soil parameters in reference site for the 10 cm increments? All the results presented in the results section concerned the 40 first cm of soil. The reference site was sampled 'until bedrock was reached (i.e., 0-5, 5-10, 10-15, 15-20cm). '(I.119-120). Does it mean that the number of sample by 10cm increment in reference site is not constant? If the bedrock can be reached at 20cm within the reference site, what are the implications for the olive orchard especially in eroded areas? What are the implications on the rock fragment content in the samples and the computation of the Corg stocks?"

The sampling in the reference area was done manually at 5 cm depth intervals (e.g. 0-5 and 5-10 cm) and the samples were integrated to perform the analysis at 10 cm intervals (e.g. integrating 0-5 and 5-10 cm into one for 0-10 cm). We found a mistake in lines 119-120; the

reference area was sampled until reaching bedrock which was located at least at 40 cm in all cases. The carbon,  $\delta$ 13C and  $\delta$ 15N analysis comes from samples from four of these pits, while the 137Cs analysis comes from the 13 pits. In all these pits bedrock was below 40 cm depth. Carbon stock calculations were made for the fine soil fraction (< 2 mm) after discounting rock or stone fragments larger than 2 mm and considering soil bulk density measured using the hand cylindrical core sampler with a volume of 100 cm3. All these have been revised in lines 121-131, and 175-176, with changes marked in red.

**"Comment 3: Could you specify somewhere what are the final numbers of values analysed by 10cm increments in the reference site and in the olive orchard please?"**

Yes. In the revised version of the text we have included the information requested by the reviewer. The number of soil samples for each 10 cm increments was 4 and 8 from the reference area and olive orchard, respectively. It is in lines 134-135 of the manuscript.

"Comment 4: The sampling was performed by a mechanical soil core. Was it a percussion drilling machine? Was there any soil deflection/compaction of the samples due to the mechanical drilling, i.e. was there any consequence on the depths of the soil increments?"

The mechanical soil sampling was made with a hydraulic core sample which gently rotates and push the core, and with the soil at a moisture content between 40 to 80 WHC, and therefore we did no need to hard drilling the soil, minimizing the compression of the soil samples. The sampling was made properly, insuring that the whole sample was taken for each given depth, abandoning the point and starting a new one if some problem arose (like a sample being only partially taken). A better explanation of this and the model of core equipment used have been included in the revised version of the manuscript, lines 121-130.

"Comment 5: The Corg stocks were calculated in the study. How exactly? Did you assess the soil bulk density based on the volume and mass of the soil increments? What about the rock fragments?"

Soil carbon stock were calculated for the fine soil fraction after discounting rock or stone fragments larger than 2 mm, and considering bulk density, and it has been clarified in lines 176-178 of the revised manuscript.

"Comment §2.3 'Physico-chemical analysis': Corg concentration were determined according to Walkley and Black method. Did you apply a coefficient of correction to the raw data in order to take into account for the incomplete oxidation? This correction factor may vary from 1 to 1.6 depending on land use, soil texture, organic matter quality, sampling depth or climate. You compare two sites with different land uses, texture and organic matter quality (as highlighted by the fractionation results), and different depths."

We thanks to the reviewers for this comment, as we have detected a mistake in the reference we used regarding the method for SOC determination. In all cases, SOC fractions and in the bulk soil, organic carbon concentrations were determined by using the wet oxidation sulfuric acid and potassium dichromate method of Anderson and Ingram (1993). We have corrected this in the revised version of the manuscript, lines 159-161.

"Comment 6: You determined the theoretical values of stable carbon saturation based on the soil particle analysis. Could you specify exactly which model you used, with the values of the parameters, please? (See my comments below concerning the results section)."

See answer to this concern in result section below Comment 200-204.

"Comment §3 'Results': *I.* 197-199: A more correct way to compare soil Corg stocks between different land uses is on equivalent soil mass."

We have compared the reference area and the olive orchard in equivalent soil mass following the procedure describe in Wend and Hauser, 2013. An equivalent soil mass procedure for monitoring soil organic carbon in multiple soil layers. European Journal of Soil Science doi: 10.1111/ejss.12002, 2013. This appears in the revised Figure 7 and in lines 176-178 and 221-225 of the revised manuscript.

"Comment. 200-204: did you invert in the values of texture distribution between the reference and olive orchard sites? If you have estimated the theoretical values of stable carbon saturation based on the content of particles <2 $\mu$ m (l. 205), the olive orchard should have a higher potential than the reference site according to the clay contents proposed here, i.e. 41 and 30% in the orchard and reference site respectively. Concerning the values of theoretical stable carbon saturation, could you precise the model used to compute them please? The values you proposed (i.e., 1.94 and 1.15%C; l.205) can't be achieved based on the model a proposed by Hassink & Whitmore (1997) in the Table 4."

We thank the reviewer for his comments on this topic as we have detected errors on our calculations regarding the theoretical values of stable carbon saturation. As reviewer has detected, the olive orchard soils have a higher potential than reference site. We have applied the model of Hassink & Whitmore (1997). According to this model, the theoretical value of protected SOC (g C kg soil-1) is calculated as =  $21,1 + (clay content (g kg-1 soil) \times 0,0375$ . Considering that there were not significant differences in the soil clay content along the catena in the olive grove soils, the average theoretical protected SOC (%) is  $3,63\pm0,19$ , whereas in the reference site averaged  $3,24\pm0,11$ . According to these new values, protected soil Corg in the reference site and orchard soils accounted for  $49.8\pm11.5$  % and  $20.5\pm5.2$  % of the maximum soil stable  $C_{org}$ , respectively at the topsoil. After this amended, conclusions have not changed respect to the first submitted version of the manuscript. In the revised version of the manuscript we have corrected the data: (lines 225-232).

**"Comment §4 'Discussion': I. 276: the value is 1.19 or 1.15%C as proposed line 205?"**

Reviewer is right on this issue, and we are deeply sorry on our errors on the calculations. The correct value is  $(3.64\pm0.23 \%)$  and we have amended in the revised version of the manuscript: (lines 303-305) In fact, the protected Corg concentration in the topsoil of the olive orchard in

the eroded area is about the  $18.6\pm3.9$  % of the upper limit of protected Corg ( $3.64\pm0.23$  %) according to the model of Hassink and Whitmore (1997). This has been corrected in lines 228-232 of the manuscript.

"Comment I. 278-280: here, the authors affirmed that the land degradation reduced the soil capacity for Corg stabilization. If the authors well used the model fitted by Hassink and Whitmore in 1997 ('As proposed by Hassink and Whitmore (1997), theoretical values of carbon saturation were established from the soil particle analysis' I. 158-159), they know that basically the model is in the form : X = a \* clay content + b with X the soil capacity for Corg stabilization, a and b some constants. As the soils in the reference and in the olive orchard have different clay content, they have different capacity for Corg stabilization! Here, it is like the authors were affirming that the land degradation has changed soil texture... I need more explanation and proof, please."

We agree with the reviewer. This sentence has been deleted in the revised version of the manuscript.

"Technical corrections Figure 1: Please, could you add bar scales or precise the olive orchard size in the part §2.1?"

Yes we have included scale bars for the views in the revised Figure 1.

**Reviewer 3**

"General comment: This study examines changes in selected soil properties (SOC and SOC fractions, P available P and organic N) related to soil quality and explores the application of stable isotopes as indicators of soil degradation (13C and 15N) in an Calcic Cambisol under different land uses (open Mediterranean forest and orchard) in the southwestern region of Spain. Further, authors evaluated changes in the mentioned soil properties and water stable aggregates due to soil redistribution processes comparing eroded vs depositional sites within the olive orchard (areas previously identified by 137Cs technique). Please see below some comments: "

**"Comment 1: Line 23 deposition is non degraded?"**

Yes, that is our hypothesis. We have clarified this in the revised version of the manuscript, line 24

**"Comment 2: Clarify Lines 22-25 I miss results concerning 13C"**

We have added one line concerning  $\delta$  13C results in the revised version of the manuscript, lines 20-23.

"Comment 3, Line 31 Although is a text extract with meaningful information. I suggest "which seeks to increase global soil organic matter stocks by 0.4 percent per year as a compensation for the global anthropogenic C emissions" Lines 33-34 split the paragraph into two sentences."

Yes we have edited and divided this section into two sentences in the revised version of the manuscript, lines 30 to 32.

"Comment 4: Line 41 This part seems disconnected from the previous one (soil degradation & soil quality). I suggest move this part to line 41 "Olive cultivation has been linked to severe environmental issues including the acceleration of erosion and soil degradation (e.g. Beaufoy, 2001, Scheidel and Krausmann, 2011). In fact, soil degradation is ... (Gómez, 2014)."

We agree with the comment. We have edited this section in this way which looks more straightforward, lines 41 to 43.

**"Comment 5: Line 51 what is the reason for?"**

It also combined cultivation in very steep slopes and areas of high rainfall erosivity. We have edited this sentence to include this evaluation in the revised version of the manuscript, line 50.

"Comment 6: Lines 85-89? Please rewrite to improve the readability of the text Line 85 It would be very illustrative to include the 137Cs reference value and sd Line 109 State exactly the plant species (shrubs and annual grasses)."

We have include the 137Cs reference which as reported by Mabit et al. (2012), based on 13 soil profiles investigated, the initial 137Cs fallout in this undisturbed area was evaluated at 1925  $\pm$  250 Bq m-2 (mean  $\pm$  2 standard error) with a CV of 23%, lines 82-84 of the revised documents. We have also included a list of the most common shrubs and annual grasses in the study site in lines 109-112 of the manuscript.

"Comment 7 Line 120 Specify number of soil profiles deeper than 20 cm; excavation method is diddretn than mechanical method for soil sampling? Please include type of core sampler (automatic or manual soil core sampler)."

We have revised revise the manuscript to clarify the sampling method and the number of soil profiles. In the reference area the sampling was made through manual excavation while in the olive orchard the sampling at 10 cm interval was performed using a hydraulic core sample which gently rotates and push the core. Soil moisture content was the adequate to avoid hard drilling the soil. This minimizes the compression of the samples. The sampling was made checking that the whole sample was taken for each given depth abandoning the point and starting a new one if some problem arose (like a sample being only partially taken). The bulk density values shown in Table 5 were obtained using the hand cylindrical core sampler with a volume of 100cm3.

Regarding the number of samples and depths, there is a mistake in lines 120-122. The reference area was sampled until reaching bedrock which in same case was above 60 cm. In the case of the pits used for the carbon and isotopic analysis, 4 out of 13 were used, while all the 13 pits were used for the Cs137 analysis. In all cases these 13 pits reached 40 cm depth. Therefore, for the carbon and isotopic N and C analysis there were, for each soil depth (0-10, 10-20, 20-30, 30-40 cm) four replications in the reference area and 8 replications in the olive orchard.

All the revised text appeared in lines 120-135, and 173-175, with changes marked in red.

"Comment 8: Line 125 A similar table for the two reference transects could be included (137Cs inventories since SRR are not applicable in ref site)."

We have added Table in the supplementary material indicating the127Cs inventories of all the points in the reference area as well as their coordinates, indicating the four one used in the analysis. Additionally we have included the average 137Cs inventories of the reference area in lines 83-85 of the revised document.

**"Comment 9: Line 139 with sodium polytungstate "**

This misprint will be corrected in the revised version of the manuscript, line 149.

"Comment 10: Line 145 Explain in detail acid hydrolysis procedure: acid attack (acid concentration, time, temperature) and preparation for carbon analysis. Include a reference of the method."

As we mentioned in the material and method section (Physicochemical analysis) we have applied the method of Six et al. (2002) and modified by Stewart et al. (2009). Acid hydrolysis, described by Plante et al. (2006) consisted of incubating the samples (The silt+clay-size fraction from both the density flotation of the  $53 - 250 \mu m$  fraction and the initial dispersion and physical fractionation of the  $< 53 \mu m$  fraction) at 95 oC for 16 h in 25 ml of 6 M HCl. After hydrolysis, the suspension was filtered and washed with deionized water over a glass-fiber filter. Residues were dried at 60 °C and weighed. These fractions represent the non-hydrolyzable C fractions. The hydrolysable C fractions were determined by difference between the total organic C content of the fractions and the C contents of the non-hydrolyzable fractions. We have added this information in material and method section. All has been revised in lines 155-161.

**"Comment 11; Line 163 Clarify the number of soil samples at similar soil depth and considered for statistical analysis Line 174 fractions Line 206 topsoil is 0-10 cm?"**

It has been clarified in the text, clearly stating that for the carbon and isotopic N and C analysis there were, for each soil depth (0-10, 10-20, 20-30, 30-40 cm) four replications in the reference area and 8 replications in the olive orchard. In line 206 top-soil means 0-10 cm, this will also be clarified. Lines 134-135.

**"Comment 12: Lines 212-216 This part should be extended and explained in depth."**

We have revised this part expanding this explanation, text in red in paragraph between lines 359-362.

"Comment 13: Line 294 I consider there is no evidences from results for this statement (indicate selective deposition of soil aggregates). Please revise".

We have edited this section deleting this sentence.

**Variation of soil organic carbon, stable isotopes and soil quality indicators across an eroding-deposition catena in an historical Spanish olive orchard**

José A. Gómez1, Gema Guzmán2, Arsenio Toloza3, Christian Resch3, Roberto García-Ruíz4, Lionel Mabit3

1Institute for Sustainable Agriculture-CSIC, Córdoba, Spain

2Applied Physics Dept., University of Córdoba, Spain

3Soil and Water Management and Crop Nutrition Laboratory, FAO/IAEA Agriculture & Biotechnology Laboratory, IAEA Laboratories Seibersdorf, Austria

4Animal and Plant Biology and Ecology Dept., Ecology section, Center for advance studies in olive groves and olive oils. University of Jaén, Spain

Correspondence to: José A. Gómez (joseagomez@ias.csic.es)

Abstract. This study compares the distribution of bulk soil organic carbon, its fractions (unprotected, physically, chemically 15 and biochemically protected), available P ( $P_{avail}$ ), organic nitrogen ( $N_{org}$ ) and stable isotopes ( $\delta^{15}N$  and  $\delta^{13}C$ ) signatures at four soil depths (0–10, 10–20, 20–30, 30–40 cm) between a nearby open forest reference area and an historical olive orchard (established in 1856) located in Southern Spain. In addition, these soil properties, as well as water stable aggregates ( $W_{sagg}$ ) were contrasted at eroding and deposition areas within the olive orchard, previously determined using 137Cs. SOC stock in the olive orchard (about 40 t C ha-1) was only 25 % of that in the forested area (about 160 t C ha-1) at the top 40 cm of soil,

- 20 and reduction was especially severe in the unprotected organic carbon. The reference and the orchard soils also showed significant differences in the  $\delta$  13C and  $\delta$  15N signals, likely due to the different vegetation composition and N dynamics in both areas. Soil properties along a catena, from erosion to deposition areas within the old olive orchard, showed large differences. Soil Corg, Pavail and Norg contents and  $\delta$  15N at the deposition (presumably non-degraded) were significantly higher than that of the erosion area, defining two distinct areas with a different soil quality status (non-degraded vs
- 25 degraded). These overall results indicate that proper understanding of  $C_{org}$  content and soil quality in olive orchards require the consideration of the spatial variability induced by erosion/deposition processes for a convenient appraisal at farm scale.

**1** Introduction**

Research on soil organic carbon (SOC) and its dynamics has increased after the declaration of 4 per thousand program (Lal,

30 2015), which seeks to increase global soil organic carbon stocks by 0.4 percent per year as a compensation for global / anthropogenic C emissions. Under this program, special emphasis is given to combat soil degradation due to the strong

**Comentario [JA1]:** Added to address comment 2 by R3. "Comment 2: *Clarify Lines 22-25 I miss results concerning 13C*" We have added one line concerning δ 13C results in the revised version of the manuscript, lines 20-23.

Comentario [JA2]: Added to address Comment 1 by R3. "Comment 1: Line 23 deposition is non degraded?" Yes, that is our hypothesis. We have clarified this in the revised version of the manuscript, line 24

**Comentario [JA3]:** Added to address comment 3 by R3. "Comment 3, Line 31 Although is a text extract with meaningful information. I suggest "which seeks to increase global soil organic matter stocks by 0.4 percent per year as a compensation for the global anthropogenic C emissions" Lines 33-34 split the paragraph into two sentences."

Yes we have edited and divided this section into two sentences in the revised version of the manuscript, lines 30 to 32. impact on the global carbon cycle because of the depletion of the SOC stock. For instance, in European agricultural soils, Lugato et al. (2016) reported that erosion-induced SOC fluxes were in the same order as the current gains from improved management and must be reduced to maintain soil health and productivity. Lal (2003) estimated the global erosion-induced

35 displacement of SOC at 5.7 Pg C yr-1, approximately 70 % of which is redistributed and redeposited over the landscape and the remaining 30% is transported by rivers into aquatic ecosystems. SOC is the most important indicator of soil quality (Rajan et al., 2010) and erosion-induced loss of SOC affects on-site soil fertility and off-site environment quality (Lal, 2019). However, the effects of soil erosion and the fate of the specific SOC fraction transported by erosion in specific agricultural systems such as olive cropping remains poorly understood, and therefore, agro-environmental impacts of SOC dynamics and

40 variability require more site and crop specific research.

Olive trees, one of the most important crops in the Mediterranean region which account for approximately 9.7 Mha (FAOSTAT. 2019), have been linked to severe environmental impacts including the acceleration of erosion and soil degradation (e.g. Beaufoy, 2001, Scheidel and Krausmann, 2011). In fact, soil degradation is common in olive orchards as they have been traditionally cultivated under rainfed conditions on sloping land, at relatively low tree densities, limited

- 45 canopy size by pruning and bare soil management to optimize water use by the tree under the semiarid conditions which characterize the Mediterranean climate (Gómez, 2014). Indeed, there are many studies which have measured high erosion rates in olive orchards on sloping areas (e.g. Gómez et al., 2014), although these high erosion rates are not necessarily a direct consequence of current management. Vanwalleghem et al. (2011) in a study of historical erosion rates in several ancient olive orchards of Montefrío (Southern Spain) reported unsustainable erosion rates in the range of 23 to 68 Mg ha-1 y-1
- 50 during the XIX and early XX centuries, when these orchards were managed under the same slope and rainfall conditions with bare soil, albeit based on animal ploughing. Vanwalleghem et al. (2011) also reported a further increase in the erosion rates when bare soil management started to be implemented in these orchards by mechanization and herbicides, in the late XX century. In the last five decades (Ruíz de Castroviejo, 1969), there has been an attempt to control soil degradation, while maintaining a favourable soil water balance for the tree through the gradual development of temporary cover crops (grown
- 55 during the rainy season) (Gómez et al., 2014). These high erosion rates have also been linked to the degradation of soil properties observed in olive orchards. For instance, Gómez et al. (2009b) measured the differences in soil properties in a 5-year long experiment on runoff plots reporting a decrease in SOC, aggregate stability and infiltration rates in bare soil as compared to cover crop management. Such scientific evidence which links changes in soil properties to different erosion rates in olive orchards under controlled conditions are rarely reported in the literature. Indeed, most of the studies connecting
- 60 soil properties with different soil managements in olives come from surveys of soil properties in orchards placed on similar soil types but with differences in soil management. An example of these studies are those of Álvarez et al. (2010) or Soriano et al. (2014) who found an improvement in soil properties, particularly aggregate stability, SOC and biological activity, in organic olive orchards with cover crops, when compared to bare soil ones. In recent years, these studies have started to deepen our understanding in investigating key properties such as SOC. For instance, Vicente-Vicente et al. (2017) evaluated
- 65 the impact of cover crops in the distribution of unprotected and protected SOC in the top 15 cm of the soil. These field

**Comentario [JA4]:** Edited to clarify meaning as requested in annotated version of the manuscript by R1.

Comentario [JA5]: Added to address comment 4 by R3. "Comment 4: Line 41 This part seems disconnected from the previous one (soil degradation & soil quality). I suggest move this part to line 41 "Olive cultivation has been linked to severe environmental issues including the acceleration of erosion and soil degradation (e.g. Beaufoy, 2001, Scheidel and Krausmann, 2011). In fact, soil degradation is ... (Gómez, 2014)." We agree with the comment. We have edited this section in this way which looks more straightforward, lines 41 to 43.

**Comentario [JA6]:** Added to address comment 5 by R3. "Comment 5: Line 51 *what is the reason for?*" It also combined cultivation in very steep slopes and areas of high rainfall erosivity. We have edited this sentence to include this evaluation in the revised version of the manuscript, line 50. studies take samples in a representative area of the slope, which is a common assumption in many soil quality studies (e.g. Andrews and Carroll, 2001). Although there are a limited number of experiments on the spatial variability of soil properties in olive orchards, they suggest significance in-field variability (e.g. Gargouri et al., 2013; Huang et al. 2017). Moreover, Gómez et al. (2012) suggested that part of this on-site variability of soil properties, regarding organic carbon, might be related to erosion/deposition processes.

In-field variability associated with erosion/deposition processes is relatively well documented for organic carbon content in field crops (e.g., De Gryze et al. 2008, Mabit and Bernard, 1998, 2010; Van Oost et al., 2005). While the human-induced acceleration of soil erosion has depleted the SOC stock of agroecosystems, the fate of SOC transported over the landscape and that deposited in depressional sites is not fully understood, despite the fact that it might explain a high proportion of the

75 on-site variability of soil properties.

70

95

Most of the erosion rates recorded or established in olive orchards come from runoff plots or small catchment experiments (e.g. Gómez et al., 2014). The use of the 137Cs approach has demonstrated its potential in establishing long-term soil erosion rates in this specific land use. An example of these studies is that of Mabit et al. (2012) in which erosion as well as deposition rates since the 1950's were determined in one ancient olive orchard in the municipality of Montefrío, showing an

- 80 average annual rate in the eroding part of the slope of 12.3 t ha-1 yr-1, and an average deposition rate in the lower section of the hillslope, much shorter than the eroding section, of 13.1 t ha-1 yr-1. This study involved a reference area for establishing precisely the initial 137Cs inventory, a natural undisturbed area located at 200 m from the orchard. As reported by Mabit et al. (2012), based on 13 investigated soil profiles, the initial 137Cs fallout in this undisturbed area was evaluated at 1925 ± 250 Bq m-2 (mean ± 2 standard error) with a CV of 23%.
- 85 To complement and/or to circumvent some limitation associated with the use of this anthropogenic radioisotope (see Mabit et al., 2008) and to maintain the capacity to determinate erosion and deposition rates without the need to use direct measurements, other natural radioisotopes such as 210Pb (e.g. Mabit et al., 2014; Matisoff et al., 2014) or stable isotopes such as  $\delta^{15}$ N or  $\delta^{13}$ C (e.g. Meusburger et al., 2013) have been proposed.

In this study, we hypothesized that the contribution of the long-term erosion-deposition processes on the in-field variability

90 of soil properties in olive orchards (or other woody crops) under medium-high slope is relevant and should be taken into account when analysing the effects of specific strategies on SOC sequestration or on other soil properties. In addition, we exploited the advantage provided by the unique location of an ancient olive orchard near an undisturbed reference area and the previous information on this site from studies on historical erosion rates, to fulfil the following objectives:

1- To quantify the long-term variability in soil total organic carbon and in their different fractions, and soil quality indicators in relation to erosion and deposition areas in an historical olive orchard;

2- To evaluate these differences in relation to the reference values found in an undisturbed natural area;

3- To evaluate differences in stable isotopes ( $\delta^{13}$ C and  $\delta^{15}$ N) and explore their potential for identifying degraded areas within the olive orchard.

**Comentario [JA7]:** Added to address Comment 6 by R3. "Comment 6: Lines 85-89? Please rewrite to improve the readability of the text Line 85 It would be very illustrative to include the 137Cs reference value and sd Line 109 State exactly the plant species (shrubs and annual grasses)."

We have include the 137Cs reference which as reported by Mabit et al. (2012), based on 13 soil profiles investigated, the initial 137Cs fallout in this undisturbed area was evaluated at 1925  $\pm$  250 Bq m2 (mean  $\pm$  2 standard error) with a CV of 23 %, lines 82-84 of the revised documents. We have also included a list of the most common shrubs and annual grasses in the study site in lines 109-112 of the manuscript.

**Comentario [JA8]:** Added to address comment 8 by R3. "Comment 8: Line 125 A similar table for the two reference transects could be included (137Cs inventories since SRR are not applicable in ref site)."

We have added Table in the supplementary material indicating the127Cs inventories of all the points in the reference area as well as their coordinates, indicating the four one used in the analysis. Additionally we have included the average 137Cs inventories of the reference area in lines 83-85 of the revised document.

**2 Materials and Methods**

**100 2.1 Description of the area**

The study area is located in the municipality of Montefrío, southwestern Spain (Figure 1). The municipality extension is around 220 km2, of which 81 % is cultivated, mostly with olive trees. The climate in the region is continental Mediterranean with a long-term (1960–2018) average annual precipitation of 630 mm, a mean annual evapotranspiration of 750 mm, and a yearly average temperature of 15.2 °C. It is a mountainous area, with elevation ranging between 800–1600 m a.s.l. at the

- 105 highest point (Sierra de Parapanda). Soil sampling took place in two areas around the archaeological site "Peña de los Gitanos", where the soil is classified as Calcic Cambisol according to the FAO classification. The reference undisturbed area was inside an archaeological site (Figure 1). This undisturbed area is covered by open Mediterranean forest interspersed with shrubs and annual grasses on limestone material (calcarenites). The status of this protected site guarantees that no anthropogenic activities have impacted on it for a long period of time, approximately since the end of XVI century. This area
- 110 is covered by natural vegetation typical of the Mediterranean region, mainly bushes like *Pistacia lentiscus* and *Retama* sphaerocarpa and herbaceous species such as Anthemis arvensis, Calendula arvensis, Borago officinalis, Bracchypodium spp., Bromus spp., and Medicago spp. Combined with its flat topography, this area has the potential to allow the establishment of reference values for undisturbed soil. The area studied was an olive orchard located close to (some tens of meters) the reference area (Figure 1) which had been established in 1856. Both areas were described in detail in previous
- 115 studies on historical erosion rates in the region (Vanwalleghem et al., 2011, Mabit et al., 2012). This olive orchard is rainfed, and soil management in the decades before the sampling was based on bare soil with pruning residues (trees pruned every 2 years) being chopped and left on the soil surface. Olive trees are fertilized annually with 5 kg of 15 N-P-K, spread below the tree canopy area.

**2.2 Soil sampling**

- 120 The reference site, adjacent to the olive orchard, belongs to an archaeologically protected site and therefore it is a noncropped area excluded from any soil disturbance, Figure 1. This site was sampled at thirteen points across a transect, spaced at an average distance of 6 m, with only four of these sampling points used in this study (Figure 1). At each sampling point, the excavation method was used and based on the collection of soil samples at 5 cm increments until bedrock was reached (i.e. 0–5, 5–10, 10–15, 15–20 and when possible, 20–25, 25–30, 30–35, 35–40, 40–45, 45–50, 50–55 and 55–60 cm).
- 125 Composite soil samples at 10 cm interval were prepared at the laboratory to perform the chemical analysis of the reference area as described below.

In the olive orchard a hydraulic mechanical core sampler was used. It gently rotates and pushes an 8 cm in diameter core to sample 8 points in a 452 m long catena (Figure 1). To minimize soil disturbance, soil sampling was made in soil water content between 40 to 80 % of water holding capacity. Precautions were taken to assure that whole sample was collected for

**Comentario [JA9]: Added to address C6 by R3. "Comment 6: Lines 85-89?**

Please rewrite to improve the readability of the text Line 85 It would be very illustrative to include the 137Cs reference value and sd Line 109 State exactly the plant species (shrubs and annual grasses)." We have include the 137Cs reference which as reported by Mabit et al. (2012), based on 13 soil profiles investigated, the initial 137Cs fallout in this undisturbed area was evaluated at 1925  $\pm$  250 Bq m2 (mean  $\pm$  2 standard error) with a CV of 23 %, lines 82-84 of the revised documents. We hav

**Comentario [JA10]:** All text in red from her to line 135 added to address minor comment 2 by R1. "Authors collected 13 micro pits from reference sites and 8 pits from olive orchard sites. Then you created one or three composite samples for fraction/isotopic measurement or measured all micro pigs as repeats?" and highlighted sections in annotated version of the manuscript also by R1.

**Comentario [JA11]:** Lines in red from here to line 131 to address comment 2 by R2. "Comment §2.2 'Soil sampling'; The authors specified in the text that the reference site was sampled per 5 cm increments whereas the olive orchard was sampled per 10 cm increments. How did the authors compute values of soil parameters? All the results presented in the results section concerned the 40 first cm of soil. The reference site was sampled 'until

Comentario [JA12]: Comments in red from here to line 135 to address comment 7 by R3. "Comment 7 Line 120 Specify number of soil profiles deeper than 20 cm; excavation method is diddretn than mechanical method for soil sampling? Please include type of core sampler (automatic or manual soil core sampler)." We have revised revise the manuscript to clarify the sampling method and the number of soil profiles. In the reference area the sampling was made through manual

Comentario [JA13]: Added to address comment 4 by R2. "Comment 4: The sampling was performed by a mechanical soil core. Was it a percussion drilling machine? Was there any soil deflection/compaction of the samples due to the mechanical drilling, i.e. was there any consequence on the depths of the soil increments?" The mechanical soil sampling was made with a hydraulic core sample which gently rotates and push the core, and with the so each given depth, and at each sampling point in the orchard soil was taken at four different depths (0–10, 10–20, 20–30 and 30–40 cm).

In a previous study, soil erosion and deposition rates were determined at each sampling point, comparing the 137Cs inventory among these points and that of the undisturbed reference area (Mabit et al., 2012). The positions of all sampling points were recorded by RTK-GPS at submeter resolution (Table 1). Overall, 12 points were sampled, 4 in the reference area and 8 in the catena across the olive orchard, with all of them reaching the bedrock below 40 cm depth.

135 catena across the olive orchard, with all of them reaching the bedrock below 4

**2.3 Physicochemical analysis**

Soil samples were passed through a 2 mm sieve and homogenized, and stoniness determined as % in mass. Separation of the various soil organic carbon ( $C_{org}$ ) pools was performed by a combination of physical and chemical fractionation techniques through a three-step process developed by Six et al. (2002) and modified by Stewart et al. (2009), summarized here. This

- 140 three-step process isolates a total of 12 fractions and it is based on the assumed link between the isolated fractions and the protection mechanisms involved in the stabilization of organic C. First a partial dispersion and physical fractionation of the soil is performed to obtain three size fractions: >250 mm (coarse non-protected particulate organic matter, POM), 53–250 mm (microaggregate fraction), and <53 mm (easily dispersed silt and clay). This physical fractionation is done on air-dried 2-mm soil sieved over a 250-mm sieve. Material greater than 250 mm remained on the sieve. Microaggregates were
- 145 collected on a 53-mm sieve that was subsequently wet-sieved to separate the easily dispersed silt- and clay-sized fractions from the water-stable microaggregates. The suspension was centrifuged at 127 x g for 7 min to separate the silt-sized fraction. This supernatant was subsequently separated, flocculated and centrifuged at 1730 x g for 15 min to separate the clay-sized fraction. All fractions were dried in a 60 °C oven and weighed. Afterwards there was a second step involving a further fractionation of the microaggregate fraction isolated in the first step. A density flotation with sodium polytungstate
- 150 was used to isolate fine non-protected POM (LF): After removing the fine non-protected POM, the heavy fraction was dispersed overnight by shaking and passed through a 53 mm sieve to separate the microaggregate-protected POM (>53 mm in size, iPOM) from the microaggregate-derived silt and clay-sized fraction. The resulting suspension was centrifuged to separate the microaggregate-derived silt- versus clay-sized fraction as described above. A final third step involved the acid hydrolysis of each of the isolated silt- and clay-sized fractions. The silt- and clay-sized fractions from both the density
- 155 flotation and the initial dispersion and physical fractionation were subjected to acid hydrolysis. The unprotected pool includes the POM and LF fractions, isolated in the first and second fractionation steps, respectively. The physically protected SOC consists of the SOC measured in the microaggregates. It includes not only the iPOM but also the hydrolysable and non-hydrolysable SOC of the intermediate fraction (53–250 µm). The chemically and biochemically protected pools correspond to the hydrolysable and non-hydrolysable SOC in the fine fraction (< 53 µm), respectively. In all cases, SOC fractions and in
- 160 the bulk soil, organic carbon concentrations were determined by using the wet oxidation sulfuric acid and potassium dichromate method of Anderson and Ingram (1993).

**Comentario [JA14]: Added to address comment 3 by R2. Comment 3: Could you specify somewhere what are the final numbers of values analysed by 10cm increments in the reference site and in the olive orchard please?" Yes. In the revised version of the text we have included the information requested by the reviewer. The number of soil samples for each 10 cm increments was 4 and 8 from the reference area and olive orchard, respectively. It is in lines 134-135 of the manuscript. Also comment 11 by R3on the same question.**

Comentario [JA15]: Added to address Minor comment 1 by R1. "How did you define unprotected, physically, chemically and biochemistry protected C? POM is unprotected C, iPOM physically protected C? Please clarify in Material & Method." Also in annotated version of the manuscript by R1

Comentario [JA16]: Added to address comment 9 by R3. "Comment 9: Line 139 with sodium polytungstate " This misprint will be corrected in the revised version of the manuscript, line 149.

**Comentario [JA17]:** Added to address comment 10 by R3. "Comment 10: Line 145 Explain in detail acid hydrolysis procedure: acid attack (acid concentration, time, temperature) and preparation for carbon analysis. Include a reference of the method."

Comentario [JA18]: Added to address Minor comment 1 by R1. "How did you define unprotected, physically, chemically and biochemistry protected C? POM is unprotected C, iPOM physically protected C? Please clarify in Material & Method." Also in annotated version of the manuscript by R1

Comentario [JA19]: Added to address Minor comment 1 by R1. "How did you define unprotected, physically, chemically and biochemistry protected C? POM is unprotected C, iPOM physically protected C? Please clarify in Material & Method." Also in annotated version of the manuscript by R1

**Comentario [JA20]:** Address to address Comment §2.3 by R2. "Comment §2.3 'Physico-chemical analysis': Corg concentration were determined according to Walkley and Black method. Did you apply a coefficient of correction to the raw data in order to take into account for the incomplete oxidation? This correction Inorganic carbon was removed prior to stable isotope analysis by acid fumigation following the method of Harris et al. (2001). Moistened subsamples were exposed to the exhalation of HCl in a desiccator overnight. Afterwards, the samples were dried at 40° C before measuring the stable isotope ratio. The N measurements were done with unacidified samples and

- 165 the stable N isotope ratios and the C and N concentrations were measured by isotope ratio mass spectrometry (Isoprime 100 coupled with an Elementar Vario Isotope Select elemental analyser; both instruments supplied by Elementar, Langenselbold, Germany). The instrumental standard deviation for  $\delta^{15}$ N is 0.16% and 0.11% for  $\delta^{13}$ C. Stable isotopes are reported as delta values  $({}^{0}/_{00})$  which are the relative differences between the isotope ratios of the samples and the isotope ratio of a reference standard.
- 170 In addition, available phosphorus (Pavail) was determined by the Olsen method (Olsen and Summers 1982) and organic nitrogen ( $N_{org}$ ) was determined by the Kjeldahl method (Stevenson, 1982). Water stable aggregates ( $W_{sagg}$ ) were measured using the method of Barthes and Roose (2002). Soil particle size distribution was determined using the hydrometer method (Bouyoucos, 1962) for the topsoil (0-10 cm) of the reference area and the olive orchard. Bulk density values were calculated for the whole profile based on the volume, soil (material finer than 2 mm) and stone contents determined from the
- 175 excavation and core sampling described above. Additionally, top soil (0-10 cm depth) soil bulk density was measured using a manual soil core sample with a volume of 100 cm3. Soil carbon stocks were calculated for the fine soil fraction after discounting rock or stone fragments larger than 2 mm, and considering bulk density, and presented on equivalent soil mass as described by Wendt and Hauser (2013). As proposed by Hassink and Whitmore (1997), theoretical values of carbon saturation were calculated from the soil particle analysis. Finally, the soil degradation index developed by Gómez et al. 180
- (2009a) was calculated from the Corg, Pavail and Wsage.

**2.4 Statistical analysis**

The overall effect of depth and area (reference site vs. olive orchard or eroded vs. deposition area within the olive orchard) were evaluated using a two factor ANOVA (p<0.05). Additionally, for some comparison at similar soil depth, values of soil properties between two different areas were assessed using a one-way ANOVA test (p<0.05). In both situations, data were

185 log-transformed when necessary to fulfil ANOVA requirements. Exploratory analysis using Principal Component Analysis (PCA) was performed in the olive orchard area using the variables and sampling depths showing significant differences in the ANOVA analysis. This PCA analysis was complemented by determining the linear correlation coefficient variables showing the highest load on the PCA analysis and erosion/deposition rates using the Pearson correlation test. The statistical software package Stata SE14.1 was used for these analyses.

**190 3 Results**

3.1 Organic carbon concentration and distribution among fractions

Comentario [JA21]: Added to address minor comment 3 by R1. "L155 Please indicate the method you measuring bulk density, which was used in table 5" also noted in annotated version of the manuscript by R1. Also to address comment 6 by R3 on the same question.

Comentario [JA22]: Added to address comment 5 by R2. "Comment 5: The Corg stocks were calculated in the study. How exactly? Did you assess the soil bulk density based on the volume and mass of the soil increments? What about the rock fragments?

Soil carbon stock were calculated for the fine soil fraction after discounting rock or stone fragments larger than 2 mm, and considering bulk density, and it has been clarified in lines 176-178 of the revised manuscript.

Comentario [JA23]: Added to address comment §3 by R2. "Comment §3 'Results': 1. 197-199: A more correct way to compare soil Corg stocks between different land uses is on equivalent soil mass."

We have compared the reference area and the olive orchard in equivalent soil mass following the procedure describe in Wend and Hauser, 2013. An equivalent soil mass procedure for monitoring soil organic carbon in multiple soil layers. European Journal of Soil Science doi: 10.1111/ejss.12002, 2013. This appears in

the revised Figure 7 and in lines 176-178 and 221-225 of the revised manuscript.

Comentario [JA24]: New PCA analysis to address general comment of R1: "data mining and interpretation of results need to dig further". Also Comment 2 by R1

Comentario [JA25]: New correlation analysis to address comment by R1 in Table 1 in the annotated version of the manuscript.

Comentario [JA26]: To address general comment of R1, "need to reorganize the way of presenting the results". Also Comment 3 by R1. Also in annotated version of the manuscript by R1.

Table 2 shows the significance of the differences in bulk soil  $C_{org}$  and the various  $C_{org}$  fractions between reference and olive orchard plots, between soil depth and due to the interaction between both (Table 2A), and the effects of erosion/deposition ratio (Table 2B). Bulk soil  $C_{org}$  is always significantly higher in the reference area as compared to the olive orchard (Table

- 195 2A and Figure 2), and this was independent of the soil sampling depth.  $C_{org}$  values on the reference site were between 2 to 5 times higher than that of the olive orchard for a given depth, with the greater differences in the top 10 cm of the soil. Soil depth has a significant effect on bulk  $C_{org}$  and  $C_{org}$  fractions, with values typically decreasing with depth in both areas.  $C_{org}$  concentrations in the unprotected, and physically, chemically and biochemically protected fractions were significantly higher in the reference site as compared to the olive orchard (Table 2A and Figure 3).  $C_{org}$  values were between 2 to 6 times higher
- 200 for the unprotected and chemically protected fractions, and between 2 to 3.5 times for the physically and biochemically protected fractions, with differences tending to decrease with the soil depth.

Within the olive orchard, there were statistically significant differences between the erosion and deposition areas (Table 2B). Higher  $C_{org}$  values (1.1 to 0.6%) were observed in the deposition area located downslope, whereas lower values (0.85 to 0.55%) were measured in the areas with net erosion in the upper and mid sections of the catena. It is worth noting that these

- 205 differences between erosion and deposition areas are detected for overall analysis using a two-way ANOVA (Tables 2A and B), although an individual analysis at each depth (Figure 2) does not detect statistically significant differences, probably due to the moderate number of replications. Significant differences between the deposition and eroding area were also found for the unprotected and the physically and chemically protected fractions (Table 2B, Figure 4). However, differences for the biochemically protected (Table 2B, Figure 4) were not significant.
- 210 The percentage distribution of SOC among fractions was also significantly different between both areas (reference vs. olive orchard), except for the biochemically protected fraction (Table 3A, Figure 5). The reference area lays up most of the  $C_{org}$  in the unprotected fraction (between 50 and 65% approximately) with no significant trend with depth (Table 3A, Figure 5), followed in relative importance by the chemically and physically protected fractions which contributed between 18–30 % and 10–20 % of the bulk soil  $C_{org}$ , respectively. The biochemically protected fraction represents a very low percentage
- 215 (between 4 to 6 %). In the olive orchard, Corg is stored predominantly in the physically and chemically protected fractions which accounted for about 38 to 27 and 34 to 28 % respectively, followed by the pool of unprotected fraction (between 22 to 32%) (Figure 5). The biochemically protected fraction represents from 11 to 4% of the organic carbon stored in the olive orchard. There are no clear differences in the organic carbon distribution among the different fractions between the erosion and deposition areas in the olive orchard, with the exception of the physically protected fractions at 10-20 cm depth (Table 11 to 42 to 42 to 42 to 43 to 45 to 45
- 220 3B and Figure 6).

**3.2. Organic carbon stock**

SOC stock in the reference area is approximately  $160 \text{ th} \text{a}^{-1}$  being significantly higher than that of the olive orchard for an equivalent mass (Figure 7), which stores between 38 and 41 t ha-1 in the eroded and deposition areas respectively. There were no significant differences between these two orchard areas. Similar results were achieved across the top 40 cm soil

**Comentario [JA27]:** To address general comment of R1, "need to reorganize the way of presenting the results". Also Comment 3 by R1. Also in annotated version of the manuscript by R1.

Comentario [JA28]: Added to address comment §3 y R2. "Comment §3 'Results': 1. 197-199: A more correct way to compare soil Corg stocks between different land uses is on equivalent soil mass." We have compared the reference area and

the live orchard in equivalent soil mass following the procedure describe in Wend and Hauser, 2013. An equivalent soil mass procedure for monitoring soil organic carbon in multiple soil layers. European Journal of Soil Science doi: 10.1111/ejss.12002, 2013. This appears in the revised Figure 7 and in lines 176-178 and 221-225 of the revised manuscript

- 225 profile. Clay, silt and sand contents of the topsoil (0–10 cm) along the catena in the olive orchard averaged 41, 37 and 22%, respectively. Variability was low (average coefficient of variation of 17%) and there were no significant changes between the erosion and deposition areas. In the reference area, the soil has an average clay, silt and sand content of 30, 31 and 39% respectively, also with a homogeneous distribution across the sampling area (coefficient of variation of 10%). According to the Hassink and Whitmore (1997) model, the percentages of organic carbon of maximum soil stable Corg are of 3.24±0.11
- and  $3.63\pm0.19$  % in the reference site and olive orchard, respectively. So, protected  $C_{org}$  in the reference and olive orchard areas account for  $49.8\pm11.5$  % and  $20.49\pm5.2$  % of the maximum soil stable  $C_{org}$ , respectively, in the topsoil.

**3.3. $\delta^{15}$ N and $\delta^{13}$ C isotopic signal**

Figure 8 and Table 4A compare stable isotope delta values between the reference site and the overall olive orchard by depth. There are significant statistical differences in  $\delta^{15}$ N,  $\delta^{13}$ C, and  $\delta^{13}$ C: $\delta^{15}$ N ratio between the two areas, although in the case of  $\delta^{15}$ N only in the top 20 cm. Soil depth had a significant effect. When comparing differences between the erosion and

235 of  $\delta^{15}$ N only in the top 20 cm. Soil depth had a significant effect. When comparing differences between the erosion and deposition areas within the olive orchard, we detected statistical significant differences only in  $\delta^{15}$ N and  $\delta^{13}$ C: $\delta^{15}$ N ratio, most marked in the top 20 cm of the soil (Figure 8, Table 4B).

**3.4. Soil quality of topsoil across the catena**

Figure 9 depicts the comparison between the Pavail, Norg, Wsagg as well as the soil degradation index (SDI, Gómez et al.
240 2009a) at the top 10 cm of the soil between the erosion and deposition areas of the olive orchard and Table 5 shows a similar comparison for Norg, Pavail and bulk density at the different soil depths. Pavail in the deposition area is much higher than that of the erosion area in the top soil and for the whole profile, whereas no significant differences in individual soil layers were found for Norg and Wsagg. SDI, which is an aggregated indicator of these three soil variables, in the eroded area it was about 3 times higher than that in the deposition area.

**245 3.5. Overall analysis of soil property variability between eroded and deposition areas in the orchard**

Table 6 shows the loads in the three first principal components (PC) of the principal component analysis (PCA) for the variables used in this analysis. More than 70% of the variance was explained by the first two PC. Soils of the eroded and deposition area were clearly separated in the space defined by the two PC (Figure 10). The variables with higher contributions in PCs 1 and 2 were related to Pavail concentration, measured in the in 0- 10 cm and also on the 0- 40 cm of the soil profile, to δ 15N, δ 13C on 10 to 30 cm soil depth, and to Corg concentration and distribution in some fractions also in soil depths between 10 to 30 cm (Table 6). Deposition area tended to show higher values in the PC1 and PC2, and there was no clear tendency in the erosion area along the catena. The linear correlation coefficient between variables and the erosion/deposition rate was rather significant, r > ±0.742 for most of the variables. Interestingly, Pavail in the whole soil layer was highly positively correlated. Figure 11 depicts the ones with the clearest correlation with erosion/deposition rates, being

**Comentario [JA30]: Added to address comment Comment §4 'Discussion' by R2. "Comment §4 'Discussion': 1. 276: the value is 1.19 or 1.15%C as proposed line 205?"**

Reviewer is right on this issue, and we are deeply sorry on our errors on the calculations. The correct value is (3.64±0.23 %) and we have amended in

**Comentario [JA31]:** To address general comment of R1, "need to reorganize the way of presenting the results". Also Comment 3 by R1. Also in annotated version of the manuscript by R1.

**Comentario [JA32]:** To address general comment of R1, "need to reorganize the way of presenting the results". Also Comment 3 by R1.

**Comentario [JA33]:** To address general comment of R1, "need to reorganize the way of presenting the results". Also Comment 3 by R1. Also in annotated version of the manuscript by R1.

**Comentario [JA34]:** New section to address general comment of R1: "data mining and interpretation of results need to dig further". Also Comment 2 by Reviewer 255 the two most robust correlations with  $P_{avail}$  concentration across the 40cm soil depth, which presented a positive correlation with deposition rates, and  $\delta^{13}$ C at 10-20 cm depth which showed a negative correlation with deposition rates.

**4 Discussion**

**4.1 Organic carbon concentration and distribution among fractions**

After approximately 175 years of contrasted land use between the undisturbed reference site and the olive orchard, bulk soil organic carbon concentration and its fractions have been dramatically reduced in the olive orchard. Current levels of  $C_{org}$ concentration in the soil profile are approximately 20–25% of the reference area covered by the natural vegetation in the area adjacent to the orchard. This ratio is similar, albeit in the lower range, of the comparison of  $C_{org}$  in topsoil among olive orchards with different managements and natural areas reported for the region (Millgroom et al., 2007). The increased soil disturbance, the lower annual rate of biomass returned to the soil and the higher erosion rate in the olive orchard explain this

- 265 difference. In both areas, the  $C_{org}$  is clearly stratified, indicating that despite the different mechanisms involved there is a periodic input of biomass from the olive trees (e.g. fall of senescence leaves and tree pruning residues) plus the annual ground vegetation. Vicente-Vicente et al. (2017) estimated this biomass contribution in the range of 1.48 to 0.56 t ha-1 yr-1. It is worth noticing that the decrease in  $C_{org}$  as compared to the natural area is much higher than the reported rates of increase in  $C_{org}$  in olive orchards using conservation agriculture (CA) techniques, such as cover crops and incorporation of organic
- 270 residues from different sources. In a meta-analysis Vicente-Vicente et al. (2016) found a response ratio (the ratio of  $C_{org}$  under CA management as compared to bare soil managed orchard) from 1.1 to 1.9 suggesting that under CA management, which combines cover crops and organic residues,  $C_{org}$  doubled as a maximum.

Combining all  $C_{org}$  data of the olive orchard, the variability was about 35% which is similar to what has been reported so far in the few studies found on soil  $C_{org}$  variability in olive orchards. For instance, Gargouri et al. (2013) indicated a 24%

- 275 coefficient of variation (CV) in a 34 ha olive orchard in Tunisia, while Huang et al. (2017) reported an average CV of 41% in a 6.2 ha olive orchard in Southern Spain. Neither of these two studies reported clear trends in the distribution of  $C_{org}$  with topography. Huang et al. (2017) pointed out the additional difficulties in the determination of  $C_{org}$ . due to the topography heterogeneity, although this was compounded by the fact that within the orchards there were two areas with different planting dates for the trees. Gómez et al. (2012) reported a CV of 49% with higher  $C_{org}$  in areas where there was a change in
- 280 the slope gradient from the hillslope to a draining central channel into the catchment, although they could not find a simple relationship between the increase in content and the topographic indexes. Despite the fact that a lot of work has been done on the correlation between erosion-deposition and the redistribution of soil  $C_{org}$ , (e.g. Van Oost et al. 2005), our study is, to our knowledge, the first attempt to quantify this in detail under olive orchard agro-environmental conditions. The variability induced by the combined effects of water and tillage erosion in this olive orchard was similar to that described in other
- agroecosystems. For instance, Van Oost et al. (2005) measured on two field crop sites under temperate climate, a clear correlation between the erosion-deposition rates and the topsoil  $C_{org}$  concentration, which ranged between 0.8 % of the

**Comentario [JA35]:** To address general comment of R1, "need to reorganize the way of presenting the results". Also Comment 3 by R1. Also in annotated version of the manuscript by R1.

erosion to 1.4% of the deposition sites in the top 25 cm of the soil. Besides this, Bameri et al. (2015), in a field crop site with a semi-arid environment, also measured a higher  $C_{org}$  in the lower part of the field where deposition of the eroded soil from the upper zones took place with a mean  $C_{org}$  value of 0.95% in the top 20 cm of the soil and a CV of 53%. Overall, under such landscapes cultivated for a long time, the cumulative effect of tillage and water erosion on the redistribution of soil across the slope has been observed (Dlugoß et al., 2012). These processes also produce a vertical redistribution of  $C_{org}$

across the slope has been observed (Dlugols et al., 2012). These processes also produce a vertical redistribution of Corg resulting in a relatively homogeneous profile in the tilled layer (top 15–20 cm) and a gradual decline below this depth, as noted in our study.

**295 4.2. Organic carbon stock**

290

The differences in soil organic carbon stock between the reference site and the olive orchard are similar to those described previously when comparing cropland and forested areas, with the latter presenting a higher concentration of  $C_{org}$ , and most of it in the unprotected fraction, while the cropland presented a higher fraction of the carbon in the physically and chemically protected fractions (e.g. Poeplau and Don, 2013). This is likely due to the fact that under soil degradation processes, such as

- 300 water erosion, and low annual organic carbon inputs, as is the case under olive orchard land use, most of the unprotected  $C_{org}$  decomposes relatively quickly and a great proportion of the remained low SOC is protected. In addition, the mobilisation of the unprotected  $C_{org}$  is expected to be reduced in the protected forested area because of the canopy and the existing vegetation on the ground that protects the soil against runoff and splash erosion processes. In fact, the protected  $C_{org}$  concentration in the topsoil of the olive orchard in the eroded area is about the 18.6±3.9 % of the upper limit of protected
- 305 Corg (3.64±0.23 %) according to the model of Hassink and Whitmore (1997). Therefore, the low unprotected SOC concentration found in the olive orchard is an issue in the increase of SOC stocks. This is because protected fractions are fuelled from recently derived, partially decomposed plant residues together with microbial and micro, meso and macrofaunal debris (unprotected organic carbon) throughout processes like SOC aggregation into macro- and/or microaggregates (physically protected SOC) and complex SOC associations with clay and silt particles (chemically protected SOC) which are
- 310 disrupted in the cropland area in comparison with the reference area. The distribution among soil  $C_{org}$  fractions in the orchard of this study was similar to the result obtained by Vicente-Vicente et al. (2017) who measured  $C_{org}$  fractions distribution in olive oil orchards with temporary cover crops, with the exception of the unprotected SOC, which was much higher in soils under cover crops than that of our study under bare soil. Also interesting is the difficulty obtaining statistically significant differences in SOC stock between the eroding and deposition area (Figure 7) despite the apparent clear differences between
- 315 the two areas in some other soil properties, such as  $P_{avail}$  or  $\delta^{15}N$  and  $\delta^{13}C$  isotopic signal in the subsoil (see below).

**4.3. $\delta^{15}$ N and $\delta^{13}$ C isotopic signal**

**Comentario [JA36]:** To address general comment of R1, "need to reorganize the way of presenting the results". Also Comment 3 by R1. Also in annotated version of the manuscript by R1.

**Comentario [JA37]:** Added to address minor comment 4 by R1. L205 "Authors mentioned that "protected Corg in the reference and olive orchard area account for 87% and 64% of maximum soil stable Corg, respectively at the topsoil", it means reference area has a higher percentage of protected SOC than that of an olive orchard. This tendency is contrary to what has shown in Fig.5. How do you explain it? Please detail the way you calculated maximum soil stable Corg in Material & Method (insert equation for example?)".

**Comentario [JA38]:** To address general comment of R1, "need to reorganize the way of presenting the results". Also Comment 3 by R1. Also in annotated version of the manuscript by R1.

Differences in vegetation types induced differences in  $\delta^{13}$ C between the olive orchard and the reference area but, as expected, no differences in  $\delta^{13}$ C were detected between the erosion and deposition areas in the olive orchard given the same

- 320 origin of vegetation derived organic matter, C3-plants. Interestingly, within the olive orchard, significant differences in  $\delta$  15N were detected between the erosion and deposition areas, especially for the top 20 cm of soils (Figure 9). This suggests the potential of using  $\delta$  15N as a variable for identifying degraded area in olive growing fields, as has been proposed for other eroding regions in the world (e.g. Meusburger et al. 2013), which might provide an alternative when other conventional or isotopic techniques are not available. Nevertheless, further studies exploring this potential are necessary in order to also
- 325 consider the influence of the N-P fertilizer modifying the  $\delta^{-15}$ N in relation to the reference area (e.g. Bateman and Kelly, 2007). The source of N in soil is multifarious and subject to a wide range of transformations that affect  $\delta^{-15}$ N signature and therefore we can only speculate on the reasons for this difference in  $\delta^{-15}$ N in a relatively homogeneous area. Bulk soil  $\delta^{-15}$ N tended to be more positive (e.g. more enriched in  $\delta^{-15}$ N) as N cycling rate increases soil microbial processes (e.g. N mineralization, nitrification and denitrification) resulting in products (e.g. nitrate, N2O, N2, NH3) depleted in 15N while the
- substrate from which they were formed becomes slightly enriched (Robison, 2001). The higher  $\delta^{-15}$ N signature of the soil at deposition location suggests that rates of processes involved in the N cycling are higher than in the erosion area and that is in accordance with the higher bulk Corg and Pavail contents and lower SDI of the deposition site. The relatively lower soil  $\delta^{-15}$ N signature at the reference site could be partially due to the input of litter N from the natural legumes and to the closed N cycling which characterize natural forest ecosystems. The trend in 15N enrichment with soil depth, as found in the reference
- site, is a common observation in forest and grassland sites, which has been related to different mechanisms, including 15N isotope discrimination during microbial N transformations, differential preservation of 15N-enriched soil organic matter components during N decomposition, and more recently, to the build-up of microbial 15N-enriched microbial necro mass (Huygens et al., 2008). However, there still remains the need for a careful calibration against an undisturbed reference site and a better understanding of the influence of different vegetation between the reference and the studied area in the change of the  $\delta$  15N signal for its further use as an additional tool to determine soil degradation.

**4.4. Soil quality of topsoil across the catena**

This horizontal distribution due to tillage and water erosion also simultaneously affected other soil properties and has been described previously in other field crops areas. For instance, De Gryze et al. (2008) described, in a field crop area under conventional tillage in Belgium, how Pavail almost doubled (22.9 vs. 12.2 mg kg-1) in the depositional area as compared to the eroding upper part. They also reported that in half of the field under conservation tillage, these differences in Pavail between the upper and lower areas of the field disappeared. We have observed in our sampled orchard a pronounced increase in topsoil Pavail in the deposition area of around 400% as compared to the eroding part of the orchard, which can be attributed to the deposition of enriched sediment coming from the upslope area. The cumulative effects of the differences in Corg, Pavail,

and the trend towards higher, although non-significant,  $W_{sagg}$  in the deposition area, resulted in two areas within the orchard

**Comentario [JA39]:** Added to address minor comment 5 by R1. "(L20 L300) authors suggested using  $\delta$ 15N as a proxy to identify degraded areas; does annual input of 5 kg N-P fertilizers play a role in the dynamic of  $\delta$ 15N?"

**Comentario [JA40]:** To address general comment of R1, "need to reorganize the way of presenting the results". Also Comment 3 by R1. Also in annotated version of the manuscript by R1.

with marked differences in soil quality: the eroded part which is within the range considered as degraded in the region (Gómez et al., (2009a) and the depositional area, representing 20% of the orchard transect length (Table 1), which is within the range of non-degraded according to the same index. Topography and sediment redistribution by erosive processes introduce a gradient of spatial variability that questions the concept of representative area when it comes to describing a

- 355 whole field. In fact, several studies (e.g. Dell and Sharpley, 2006), have suggested that the verification of compliance of environmental programs such as those related to Corg sequestration should be based preferentially, at least partially, on modelling approaches. Our results raise the need for a careful delineation of sub-areas when analysing soil quality indicators and/or SOC carbon stock within the same field unit. It also warns about the interpretation of these delineated areas in relation to specific soil properties, avoiding over simplification of processes. For instance, in our study case erosion/deposition
- 360 processes had a major impact on  $P_{avail}$  and soil quality, but the impact on  $C_{org}$  concentration and stock was moderate and extremely difficult to detect statistically using a moderate number of samples.

**4.5. Overall analysis of soil property variability between eroded and deposition areas in the orchard**

Our PCA and regression analysis confirmed the relatively high variability of  $C_{org}$  and stock in relation to other soil quality 365 indicators related to erosion/deposition processes, such as  $P_{avail}$ , commented in the previous section. While in the catena studied in this manuscript P cycle seems to be mostly driven by sediment mobilization, the  $C_{org}$  and N cycle seems to be much more complex. The moderate differences in  $C_{org}$  and the homogeneity in  $\delta^{-15}N$  and  $\delta^{-13}C$  isotopic signals between the eroding and deposition areas can be due to several processes, some of them discussed above, such as spatial variability of carbon inputs due to biomass in the plot, surface soil operations in the orchard and fertilization. We found both interesting

370 and worth exploring in future studies that significant correlations between erosion/deposition rates and  $C_{org}$ ,  $\delta^{-15}N$  and  $\delta^{-13}C$  related variables were found for samples from 10-20 and 20-30 cm, indicating that short term disturbance by surface processes can mask experimental determination of the impact of erosion deposition processes in this olive orchard for these variables.

**375 5 Conclusions**

1- The results indicate that erosion and deposition within the investigated old olive orchard have created a significant difference in soil properties along a catena, which is translated into different soil  $C_{org}$ ,  $P_{avail}$  and  $N_{org}$  contents and  $\delta^{15}N$ , and thus soil quality status.

2- This variability was smaller than that of the natural area, which indicated a severe depletion of SOC as compared to thenatural area and a redistribution of available organic carbon among the different SOC fractions.

3- The results suggest that  $\delta^{15}$ N has the potential for being used as an indicator of soil degradation, although more investigation under different agroecosystems would be required for confirming this statement at a larger scale.

Comentario [JA41]: Added to address Comment 12 by R3. "Comment 12: Lines 212-216 This part should be extended and explained in depth." We have revised this part expanding this explanation, text in red in paragraph between lines 359-362.

**Comentario [JA42]:** To address general comment of R1, "need to reorganize the way of presenting the results". Also Comment 3 by R1. Also in annotated version of the manuscript by R1.

**Comentario [JA43]:** To address general comment of R1: "data mining and interpretation of results need to dig further". Also Comment 2 by Reviewer 1

4- This research highlights that proper understanding and management of soil quality and Corg content in olive orchards require considering the on-site spatial variability induced by soil erosion/deposition processes.

**385 6 Acknowledgements**

The authors would like to thank professor Manuel González de Molina for providing valuable insight into the management of the olive orchard in the decades before sampling. Also, in particular, to the three anonymous reviewers of the manuscript version in Soil Discussion, for their care and attention to the manuscript. All their comments and indications have greatly helped to increase the quality of our work. This study was supported by the projects AGL2015-40128-C03-01 of the Spanish

390 Government, SHui (GA 773903) of the European Commission and EU–FEDER funds.

---

## Author Response (AR3)

We would like to thank to the reviewers for the careful assessment of the final details of the manuscript and their helpful suggestions to improve it.

We detail below each of the reviewer's comments (in italics) how we have addressed those in the revised version of the manuscript (in red). These changes have been marked in red in the revised version of the manuscript accompanying this letter.

**Reviewer 1:**

*A great effort has been made by authors and significantly improved the manuscript. The main revisions are: (1) clarified the methodology (2) figures are polished (3) extended data analysis and interpretation (4) reorganized the Results and Discussion section. I think the revised manuscript can be accepted for publication. Some minor comments:*

We appreciate the attention paid to this manuscript in all the stages, also in helping us to polish the clarity of the manuscript with a through edition at this stage.

*Comment 1 of reviewer 1: Discussion 4.5 partly is the repetition from above and could be merged into your prior discussion section.*

This section has been merged with previous section in discussion (4.4) and material and methods (3.4)

*Comment 2 of reviewer 1: I appreciate the authors try to dig into the dataset by applying various statistics, but Fig. 11 seems not to be useful.*

We have eliminated Figure 11 and edited the revised text to indicate the parameter values for the most robust correlations.

*Comment 3 of reviewer 1. Line 50: use Arabic numbers?*

It has been revised using Arabic numbers.

**Reviewer 2:**

*The authors have made revisions of the manuscript accordingly to the comments and suggestions in the previous review. However, please find below some remaining suggestions that I recommend the Authors to take into account:*

We appreciate the attention paid to this manuscript in all the stages, also in helping us to polish the clarity of the manuscript with a through edition at this stage.

*Comment 1 of reviewer 2: Line 24 Maintain the differentiation between non-degraded vs degraded, it is unclear for me in terms of soil processes. Please reconsider: eroded vs depositional sites (see line 246)*

It has been revised eliminating the mention to degraded and non-degraded.

*Comment 2 of reviewer 2: Line 29 Particularly, research on the sequestration of carbon in soils and the potential of agricultural soils to store carbon has increased*

This sentence has been revised according to the reviewer suggestion

*Comment 3 of reviewer 2: Line 83 the local reference 137Cs inventory calculated was 1925 … instead of the initial 13çCs fallout*

It has been revised according to the reviewer suggestion

*Comment 4 of reviewer 2: Line 93 References of these previous studies or information*

These references have been added.

*Comment 5 of reviewer 2: Line 113 establishment of a reference site. 'values for undisturbed soil' could be deleted  The meaning of this sentence is unclear. Consider rewriting 'the reference area (Figure 1) which had been established in 1856' or link with information in line 120*

We have deleted the sentence and doing this, the paragraph is clear now.

*Comment 6 of reviewer 2: Line 121 where neither erosion nor sedimentation occurs (remove excluded from any soil disturbance)*

It has been revised according to the reviewer suggestion.

*Comment 7 of reviewer 2: Line 122 criterion to exclude 9 sampling point or select only 4 sampling points*

It has been explained in the revised text.

*Comment 8 of reviewer 2: Line 124 (i.e. 0–5, 5–10, 10–15, 15–20 and when possible, 20–25, 25–30, 30–35, 35–40, 40–45, 45–50, 50–55 and 55–60 cm). delete and state clearly in line 124 that soil profiles were sectioned or divided at 5 cm depth intervals till bedrock as reached variying between 20 -60 cm?*

This sentence has been revised as indicated by the reviewer.

*Comment 9 of reviewer 2: Line 140 – 141 Please clarify in the present form this sentence is unclear*

This sentence has been edited for clarity.

*Comment 10 of reviewer 2: Line 177 BD is soil fraction <2 mm / Volume soil fraction <2 mm?*

This sentence has been edited for clarity.

*Comment 11 of reviewer 2: Line 198 include p value. Revise in the whole manuscript (e.g line 242)*

These p values have been revised and included in the manuscript.

*Comment 12 of reviewer 2: Line 226 changes in clay, silt and sand contents …*

These values have been included.

*Comment 13 of reviewer 2: Line 253 revise r=0.742*

It has been revised.

*Comment 14 of reviewer 2: Lines 358-359 This sentence is unclear*

This sentence has been edited for clarity.

---

## Author Response (AR4)

Dear Topical Editor,

We would like to thank your careful revision of the manuscript edition. We have gone, among three of the authors, through all your comments in the annotated version of the manuscript, including your suggestions and correcting any additional typo that we have detected.

All these latest changes have been left in red in the marked, non-annotated, revised version uploaded to the editorial system.

Yours sincerely,